# RACCooN: A Versatile Instructional Video Editing Framework with Auto-Generated Narratives

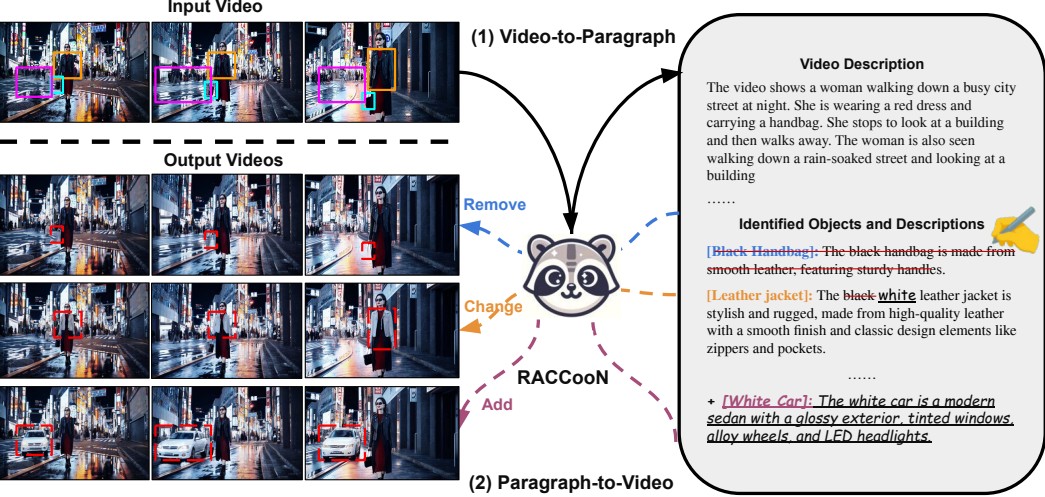

Figure 1: **Overview of RACCooN**, a versatile and user-friendly video-to-paragraph-to-video framework, enables users to remove, add, or change video content via updating auto-generated narratives.

## ABSTRACT

Recent video generative models primarily rely on carefully written text prompts for specific tasks, like inpainting or style editing. They require labor-intensive textual descriptions for input videos, hindering their flexibility to adapt personal/raw videos to user specifications. This paper proposes **RACCooN**, a versatile and user-friendly **video-to-paragraph-to-video** generative framework that supports multiple video editing capabilities such as removal, addition, and modification, through a unified pipeline. RACCooN consists of two principal stages: *Video-to-Paragraph* (V2P) and *Paragraph-to-Video* (P2V). In the V2P stage, we automatically describe video scenes in well-structured natural language, capturing both the holistic context and focused object details. Subsequently, in the P2V stage, users can optionally refine these descriptions to guide the video diffusion model, enabling various modifications to the input video, such as removing, changing subjects, and/or adding new objects. The proposed approach stands out from other methods through several significant contributions: (1) RACCooN suggests a multi-granular spatiotemporal pooling strategy to generate well-structured video descriptions, capturing both the broad context and object details without requiring complex human annotations, simplifying precise video content editing based on text for users. (2) Our video generative model incorporates auto-generated narratives or instructions to enhance the quality and accuracy of the generated content. (3) RACCooN also plans to imagine new objects in a given video, so users simply prompt the model to receive a detailed video editing plan for complex video editing. The proposed framework demonstrates impressive versatile capabilities in video-to-paragraph generation (up to $9.4\%p \uparrow$ absolute improvement in human evaluations against the baseline), video content editing (relative $49.7\% \downarrow$ in FVD), and can be incorporated into other SoTA video generative models for further enhancement.

# 1 INTRODUCTION

Recent advances in video generative models (Yan et al., 2021; Hong et al., 2022; Esser et al., 2023; Mei & Patel, 2023; Chai et al., 2023; Ceylan et al., 2023; Blattmann et al., 2023; Wu et al., 2023a), including Sora (openai, 2024), have demonstrated remarkable capabilities in creating high-quality videos. Simultaneously, video editing models (Geyer et al., 2023; Qi et al., 2023; Wang et al., 2024b; Yu et al., 2023; Wu et al., 2024; Zhang et al., 2023b) have gained significant attention, thanks to their promising applications that allow users to modify videos according to *user-written* textual instructions, effectively altering video content and attributes. Despite these advancements, significant challenges remain in developing a versatile and user-friendly framework that facilitates easy video modification for personal use. The primary challenges include: 1) the complexity of training a unified framework encompassing multiple video editing skills (e.g., remove, add, or change an object). Training a single model to perform various editing skills is highly challenging, and recent video editing methods often focus on specific tasks, such as background inpainting (Yu et al., 2023; Wu et al., 2024), or attribute editing (Geyer et al., 2023; Qi et al., 2023; Jeong & Ye, 2023). 2) the necessity for well-structured textual prompts that accurately describe videos and can be edited to support diverse video editing skills. The quality of prompts critically influences the models' capabilities and the quality of their outputs. Generating detailed prompts is time-consuming and costly, and the quality varies depending on the expertise of the annotators. Although Multimodal Large Language Models (MLLMs) (Liu et al., 2023b; Munasinghe et al., 2023; Yang et al., 2023; Yu et al., 2024) have been explored for automatically describing videos, they often overlook critical details in complex scenes. This oversight compromises the development of a seamless pipeline, hindering both user convenience and the effectiveness of video generative models.

To tackle these limitations, as shown in Fig. 1, we introduce **RACCooN**: *A Versatile Instructional Video Editing Framework with Auto-Generated Narratives*, a novel **video-to-paragraph-to-video (V2P2V)** generative framework that facilitates diverse video editing (Remove, Add, and Change) capabilities based on auto-generated narratives. RACCooN allows for the seamless removal and modification of subject attributes, as well as the addition of new objects to videos **without requiring densely annotated video prompts or extensive user planning**. Our framework operates in two main stages: *video-to-paragraph* (V2P) and *paragraph-to-video* (P2V). In the V2P stage, we introduce a new video descriptive framework built on a pre-trained Video-LLM backbone (PG-Video-LLaVA (Munasinghe et al., 2023)). We find that existing Video-LLMs effectively capture holistic video features, yet often overlook detailed cues that are critical for accurate video editing, as users may be interested in altering these missing contexts. To address this, we propose a novel multi-granular video perception strategy that leverages superpixels (Li & Chen, 2015; Ke et al., 2023) to capture diverse and informative localized contexts throughout a video. We first extract fine-grained superpixels using a lightweight predictor (Yang et al., 2020) and then apply overlapping k-means clustering (Cleuziou, 2007; Whang et al., 2015; Khanmohammadi et al., 2017) to segment visual scenes into various levels of granularity. The suggested localized spatiotemporal segmentation assists the LLM's comprehension of objects, actions, and events within the video, enabling it to generate fluent and detailed natural language descriptions. Next, in the P2V stage, to integrate multiple editing capabilities into a single model, we fine-tuned a video inpainting model that can paint video objects accurately with detailed text, object masks, and condition video. Then, by utilizing user-modified prompts from generated descriptions in the V2P stage, our video diffusion model can accurately *paint* corresponding video regions, ensuring that textual updates from prompts are reflected in various editing tasks. Moreover, to better support our model training, we have collected the **V**ideo **P**aragraph with **L**ocalized **M**ask (**VPLM**) dataset—a collection of over 7.2K high-quality video-paragraph descriptions and 5.5k detailed object descriptions with masks, annotated from the publicly available dataset using GPT-4V (Achiam et al., 2023).

We emphasize that RACCooN enhances the quality and versatility of video editing by leveraging detailed, automatically generated textual prompts that minimize ambiguity and refine the scope of generation. We validate the extensive capabilities of the RACCooN framework in both V2P generation, text-based video content editing, and video generation on ActivityNet (Krishna et al., 2017), YouCook2 (Zhou et al., 2018a), UCF101 (Soomro et al., 2012), DAVIS (Pont-Tuset et al., 2017), and our proposed VPLM datasets. On the V2P side, RACCooN outperforms several strong video captioning baselines (Li et al., 2023; Munasinghe et al., 2023; Liu et al., 2023b), particularly improving by average **+9.1%p** on VPLM and up to **+9.4%p** on YouCook2 compared to PG-VL (Munasinghe et al., 2023), based on both automatic metrics and human evaluation. On the P2V side, RACCooN

surpasses previous strong video editing/inpainting baselines (Geyer et al., 2023; Qi et al., 2023; Wang et al., 2024b; Yu et al., 2023; Wu et al., 2024) over three subtasks of video content editing (remove, add, and change video objects) over 9 metrics. We also demonstrate that the proposed RACCooN framework can enhance SoTA video generative models by leveraging detailed auto-generated textual prompts. We further conduct extensive ablation and visualizations to validate the improvement quantitatively and qualitatively. Our contributions are as follows:

1. **Framework Contribution:** RACCooN offers a user-friendly and unified framework for various video editing tasks. It provides improved interpretability and interactive experiences by automatically generating detailed, object-centric video descriptions and layout plans tailored to different editing objectives, which cannot be done through a simple combination of existing models.

2. **Technical Contribution:** We present a novel **multi-granular pooling** strategy to capture local video contexts, enhancing video comprehension by generating fluent and detailed descriptions in a zero-shot setting. This enables users to create new videos that retain the visual characteristics of the input and focus on specific context editing.

3. **Training/Dataset Contribution:** To enable RACCooN to follow complex and varied user requests for video editing, we present the **VPLM** dataset, which contains 7.2K high-quality detailed video paragraphs and 5.5K object-level detailed caption-mask pairs. This dataset facilitates accurate V2P and P2V stages with high-quality, localized textural prompts and videos.

## 2 RELATED WORK

**Video-to-Paragraph Generation.** The recent trend in video-language tasks focuses on generating comprehensive textual descriptions for long and complex video content (Shen et al., 2017; Krishna et al., 2017; Wang et al., 2018; Tewel et al., 2022; Wu et al., 2023b). Vid2Seq (Yang et al., 2023) introduces a novel dense event captioning approach for narrated videos, with time tokens and event boundaries. Video-LLaVA variants (Lin et al., 2023a; Munasinghe et al., 2023) present a large multimodal model integrating text, video, and audio inputs for generative and question-answering tasks. Similarly, LLaVA-Next (Zhang et al., 2024) improves zero-shot video understanding by transferring multi-image knowledge through concatenated visual tokens. While these methods are effective in video description, they often miss key contextual details (Zhang et al., 2023a; Li et al., 2023). Our RACCooN captures both holistic and localized details by leveraging localized spatiotemporal information, enhancing video editing and generation capabilities.

**Prompt-to-Video Editing.** Video editing (Ceylan et al., 2023; Liu et al., 2023c; Couairon et al., 2023; Kondratyuk et al., 2023; Wang et al., 2023; Zhang et al., 2023c) involves enhancing, modifying, or manipulating video content for desired effects. VideoComposer (Wang et al., 2024b) offers a multi-source controllable video generative framework. TokenFlow (Geyer et al., 2023) adapts text-to-image diffusion with flow matching for consistent text-driven video editing. LGVI (Wu et al., 2024) integrates an MLLM for complex language-based video inpainting. These methods often focus on specific tasks and may inadvertently alter unrelated regions due to limited contextual information. Our V2P2V framework overcomes these limitations by using auto-generated, detailed descriptions to integrate key contexts into diverse editing tasks.

## 3 RACCOON: A VERSATILE INSTRUCTIONAL VIDEO EDITING FRAMEWORK WITH AUTO-GENERATED NARRATIVES

Conditional video generation and editing models struggle with complex scenes due to vague text descriptions and limited video understanding. Despite improvements from recent advances in MLLMs, these models still struggle to capture complex spatial-temporal dynamics, often omitting crucial objects and details. Training a text-to-video model with such vague prompts compromises output specificity, and leads the model to generate average, arbitrary content that fails to capture user instructions' nuances. To address these issues, we introduce RACCooN, a user-friendly, two-stage video-to-paragraph-to-video editing framework. Initially, RACCooN generates detailed, structured paragraphs from videos, capturing holistic content and key local objects through multi-granular spatiotemporal information. These detailed descriptions are then used for conditional video generation and editing, enabling users to add, remove, or change video objects directly by interacting with the generated descriptions, thus enhancing the specificity and relevance of the output.

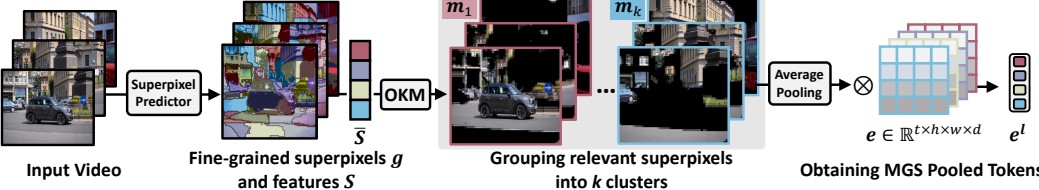

Figure 2: **Illustration of RACCooN framework.** RACCooN generates video descriptions with the three distinct pooled visual tokens, including Multi-Granular Spatiotemporal (MGS) Pooling. Next, users can edit the generated descriptions by adding, removing, or modifying words to create new videos. Note that for adding object tasks, if users do not provide layout information for the objects they want to add, RACCooN can predict the target layout in each frame.

## 3.1 V2P: AUTO-DESCRIPTIVE FRAMEWORK WITH MULTI-GRANULAR VIDEO PERCEPTION

**Multimodal LLM for Video Paragraph Generation.** In the V2P stage, the RACCooN framework generates well-structured, detailed descriptions for both holistic videos and local objects. It employs a multimodal LLM with three main components: a visual encoder $E$, a multimodal projector, and an LLM. Given an input video $\boldsymbol{x} \in \mathbb{R}^{F \times C \times H \times W}$, where $F$, $C$, $H$, and $W$ represent the number of frames, channels, height, and width, respectively, we extract video features using the visual encoder: $\boldsymbol{e} = E(\boldsymbol{x}) \in \mathbb{R}^{t \times h \times w \times d}$. Here, $t$, $h$, $w$, and $d$ denote the encoded temporal dimension, the height and width of the tokens, and the feature dimension. To understand complex videos with multiple scenes, we use three pooling strategies: *spatial pooling*, *temporal pooling*, and *multi-granular spatiotemporal pooling*. Spatial pooling $\boldsymbol{e}^s = \text{Pooling}^s(\boldsymbol{e}) \in \mathbb{R}^{t \times d}$ aggregates tokens within the same frame, while temporal pooling $\boldsymbol{e}^t = \text{Pooling}^t(\boldsymbol{e}) \in \mathbb{R}^{(h \cdot w) \times d}$ averages features across the temporal dimension for the same region. Despite these strategies helping the LLM grasp the video holistically in space or time, they often overlook capturing key objects or actions localized throughout the video stream, especially in untrimmed, dynamic, multi-scene videos.

Figure 3: **Illustration of MGS pooling.** We obtain MGS pooling tokens using a spatiotemporal mask $\boldsymbol{m}$ via overlapping k-means clustering (OKM) of averaged superpixel features $\bar{\boldsymbol{S}}$.

**Multi-Granular Spatiotemporal Pooling.** To address this issue, we introduce a novel superpixel-based spatiotemporal pooling strategy, coined *multi-granular spatiotemporal pooling* (MGS pooling). As illustrated in Fig. 2 left top, this strategy is designed to capture localized information via superpixels across spatial and temporal dimensions. Superpixels (Li & Chen, 2015; Giordano et al., 2015; Yang et al., 2020; Ke et al., 2023) are small and coherent clusters of pixels that share similar characteristics, such as color or texture. These clusters provide an efficient representation of visual scenes and are resilient to frame noise since they average out the pixel values within each cluster, effectively smoothing out variations induced by noise. As shown in Fig. 3, we use a lightweight superpixel predictor $\sigma(\cdot)$ (Yang et al., 2020) to generate superpixels across video frames, capturing the granular visuality of each local area. However, due to their limited coverage area, these fine-grained visual

features often fail to capture attribute-level semantics, such as objects and actions (Zhang et al., 2023a; Li et al., 2023). Motivated by the importance of varying the compositions of multiple superpixels for different contexts in video understanding, we propose the use of overlapping k-means clustering (Cleuziou, 2007; Whang et al., 2015) for the obtained video superpixels, which improves the granularity from fine to coarse. This approach allows the LLM to gather informative cues about various objects and actions. We first obtain the pixel features and the superpixel index vector for the video pixels: $\boldsymbol{S}, \boldsymbol{g} = \sigma(\boldsymbol{x}, \boldsymbol{g}_{\text{init}})$, where $\boldsymbol{g}_{\text{init}}$ is the input superpixel indices, initialized by a region-based grid. Given the averaged pixel features of each superpixel, $\bar{\boldsymbol{S}} \in \mathbb{R}^{|\boldsymbol{g}| \times d_p}$, where $d_s$ denotes the pixel feature size, we generate the MGS tokens $\boldsymbol{e}^l$:

$$\begin{aligned}
\boldsymbol{m} &= \text{OKM}\left(\bar{\boldsymbol{S}}, k, v\right) \in \{0, 1\}^{k \times F \times H \times W}, \\
\boldsymbol{e}^l &= \text{AvgPool}(\boldsymbol{m}) \otimes \boldsymbol{e} \in \mathbb{R}^{k \times d},
\end{aligned} \tag{1}$$

where OKM represents the overlapping k-means algorithm with $k$ centroids and overlap scale $v$ for each cluster. $\boldsymbol{m}$ denotes the set of binary masks for superpixels. $\otimes$ denotes tensor multiplication. We describe the detailed MGS process and ablation of pooling strategies in the Appendix. Next, we concatenate the pooled video tokens and map them into the text embedding space using the multimodal linear projector. Combined with the embedding of the encoded text token $\boldsymbol{e}^p$ from the textual prompt, the LLM generates a well-structured and detailed description $\boldsymbol{a}$ of the video:

$$\widehat{\boldsymbol{e}} = \text{CONCAT}[\boldsymbol{e}^s; \boldsymbol{e}^l; \boldsymbol{e}^t] \cdot \boldsymbol{W}^\top, \qquad \boldsymbol{a} = LLM(\text{CONCAT}[\boldsymbol{e}^p; \widehat{\boldsymbol{e}}]), \tag{2}$$

where $\boldsymbol{W} \in \mathbb{R}^{d \times d'}$ is the weight matrix for linear projection into the text embedding dimension $d'$. We highlight that our video description framework serves as an integrated, user-interactive tool for video-to-paragraph generation and video content editing. (Fig. 2 top right).

## 3.2 P2V: USER-INTERACTIVE VIDEO EDITING WITH AUTO-GENERATED DESCRIPTIONS

With the well-structured, detailed, and object-centric video description generated from the *Video-to-Paragraph* stage, users can 'read' the video details and interactively modify the content by altering the model-generated description. This approach shifts users' focus from labor-intensive video observation to content editing. We categorize general video content editing into three important subtasks: **(1) Video Object Adding:** add extra objects to a video. **(2) Video Object Removing:** delete target objects and re-generate the object region as the background. **(3) Video Object Changing:** change objects' attributes (e.g., color, textural, material). Many previous works have made great progress in video editing (Wu et al., 2024; Geyer et al., 2023; Qi et al., 2023; Zhang et al., 2023b; Fan et al., 2024) but usually focus on one of these subtasks. In this paper, we propose a unified generative model for video content editing that integrates all those crucial subtasks. Specifically, we formulate these subtasks as text-based video painting tasks and leverage a single video diffusion model for adding, removing, and changing video objects in the form of inpainting.

As shown in Fig. 2 bottom, our video diffusion model processes input video $\boldsymbol{x} \in \mathbb{R}^{F \times C \times H \times W}$ with a predicted binary mask $\boldsymbol{m}' \in \mathbb{R}^{F \times 1 \times H \times W}$ targeting specific regions for modification. Following image inpainting techniques (Xie et al., 2023; Rombach et al., 2022), we apply the mask[1] to the video to designate the editing region. The masked video is then encoded using a Variational Autoencoder (VAE (Kingma & Welling, 2013)) to serve as the generation condition. The model can then be informed on which video region should be edited for localized editing. Driven by the detailed description, the diffusion model can conduct diverse video editing that reflects the text prompts.

In addition, we provide details regarding the process of adding objects in video editing. Indeed, adding objects can be considered a unique video editing task, distinct from removing objects or changing attributes. Unlike the latter scenarios, where the target objects are already present in the initial video, adding objects involves introducing entirely new elements, which necessitates a slightly different editing process.

As illustrated in Fig. 2, the MLLM in the V2P process provides not only detailed descriptions but also frame-wise placement suggestions for new objects in the form of bounding box sequences. The object insertion process in RACCooN in inference is conducted through the following steps:

---

[1]We use image grounding (Liu et al., 2023d) and video tracking models (Cheng et al., 2023) as the off-the-shelf mask predictor in inference.

*1. User Edit:* The user provides an instruction to add a specific object.

*2. MLLM Output:* The finetuned MLLM in V2P generates fine-grained video descriptions along with frame-wise bounding box suggestions for new objects. For example, "Layouts of <Obj> to be added: {Frame 1: [0.2, 0.0, 0.5, 0.7], Frame 2: [0.2, 0.1, 0.4, 0.65], ... }" specifies the layout for each frame, where [x1, y1, x2, y2] represents the top-left and bottom-right corners of the bounding box, with coordinates normalized to the range $[0, 1]$ (the yellow box in Fig. 2, top right).

*3. Video editing:* Generate videos based on the MLLM-generated output, including the frame-wise layout of the object to be added.

### 3.3 VPLM DATASET COLLECTION AND RACCooN PIPELINE TRAINING

**Dataset Collection.** We utilize video datasets (Majumdar et al., 2020; Gavrilyuk et al., 2018) from previous video inpainting work (Wu et al., 2024). Each raw video is accompanied by multiple inpainted versions with specific objects removed and includes binary masks of these objects. Although well-annotated with object masks and inpainted backgrounds, these datasets lack detailed descriptions of holistic video and specific local objects, hindering RACCooN's training for producing well-structured captions for video editing. To address this, we use GPT-4V (Achiam et al., 2023) to annotate detailed video descriptions. We first re-arrange uniformly sampled video frames into a grid-image (Fan et al., 2021) and add visual prompts by numbering each frame. We then ask GPT-4V to generate detailed captions for both the entire video and key objects, in a well-structured format. Next, we train V2P and P2V stages in our framework separately (Fig. 2). In the end, RACCooN can automatically generate detailed, well-structured descriptions for raw videos and adapt these descriptions based on user updates for various video content editing tasks.

**MLLM Instructional Fine-tuning.** To enable the MLLM to output detailed video descriptions for content editing, we construct an instructional fine-tuning dataset based on VPLM with two video-instruction (Liu et al., 2023b) designs: (1) For object editing and removal, the MLLM generates structured video captions identifying key objects in the original video $x$, using annotated descriptions as the learning objective. This allows users to edit videos directly from these descriptions without exhaustive analysis. (2) For object insertion, the MLLM provides not only detailed descriptions but also frame-wise placement suggestions for new objects, enhancing its utility in video editing by avoiding manual trajectory outlining. For training, we convert video object segmentation masks into bounding boxes by selecting maximal and minimal coordinates and follow the box planning strategy using LLMs (Lin et al., 2023b). We input box coordinates as a sequence of numbers and train RACCooN framework to predict these layouts given inpainted videos $\hat{x}$. We perform parameter-efficient fine-tuning with LoRA (Hu et al., 2022)[2] on these mixed datasets with CE loss. We freeze the visual encoder and LLM backbone, updating the projector, LoRA, and LLM head.

**Video Diffusion Model Fine-tuning.** Our video diffusion model builds on the prior image inpainting model (Rombach et al., 2022), enhanced with temporal attention layers to capture video dynamics. The model is designed to generate video that aligns with input prompts, focusing on object-centric video content editing. To support this, we develop a training dataset of mask-object-description triples. We use GPT-4 to produce single-object descriptions from long, detailed video narratives, framing this task as a multi-choice QA problem. Next, for the three video editing subtasks, we design specific input-output combinations: (1) **Video Object Addition:** Inputs: inpainted video $\hat{x}$, object bounding boxes from segmentation masks $m$, and detailed object description $p$. Output: original video $x$. (2) **Video Object Removal:** Inputs: original video $x$, object segmentation masks $m$, and a fixed background prompt. Output: inpainted video $\hat{x}$. (3) **Video Object Change:** Inputs: original video $x$, object segmentation masks $m$, and object description $p$. Output: original video $x$. The model is fine-tuned following the prior work (Wu et al., 2023a), updating only the temporal layers and the query projections within the self-attention and cross-attention modules. We employ the MSE loss between generated and random noise. See Appendix for more details on the dataset and training.

## 4 EXPERIMENTAL RESULTS

**Tasks & Datasets**: We evaluate our RACCooN framework on diverse video datasets across tasks, including video captioning (**YouCook2** (Zhou et al., 2018a), **VPLM**), text-based video content editing

---

[2]We employ LoRA for query and value for each self-attention.

Table 1: **Results of Single Object Prediction** on VPLM test set. Metrics are abbreviated:**S**: *SPICE*, **B**: *BLEU-4*, **C**: *CIDEr*.

| Methods | S | B | C | IoU | FVD | CLIP |
|---|---|---|---|---|---|---|
| *open-source MLLMs* | | | | | | |
| LLaVA (Liu et al., 2023b) | 17.4 | 27.5 | 18.5 | - | - | - |
| Video-Chat (Li et al., 2023) | 18.2 | 25.3 | 19.1 | - | - | - |
| PG-VL (Munasinghe et al., 2023) | 18.2 | 27.4 | 14.6 | - | - | - |
| *proprietary MLLMs* | | | | | | |
| Gemini 1.5 Pro (Team et al., 2023) | 19.2 | 23.5 | 11.0 | 0.115 | **371.63** | 0.978 |
| GPT-4o (gpt 4o, 2024) | 20.6 | 28.0 | **37.4** | 0.179 | 447.67 | 0.977 |
| **RACCooN** | **23.1** | **31.0** | 33.5 | **0.218** | 432.42 | **0.983** |

Table 2: **Results of Human Evaluation** on YouCook2. We measure the quality of the description through four metrics: Logic Fluency (Logic), Language Fluency (Lang.), Video Summary (Summ.), and Video Details (Details). We report the normalized score $s \in [0, 100]$.

| Methods | Logic | Lang. | Summ. | Details | Avg. |
|---|---|---|---|---|---|
| Ground Truth | 66.7 | 42.2 | 41.7 | **72.2** | 55.7 |
| PG-VL (Munasinghe et al., 2023) | 77.2 | 81.1 | 69.4 | 62.8 | 72.6 |
| **RACCooN** | **80.6** | **85.0** | **72.2** | **72.2** | **77.5** |

**Input Video**

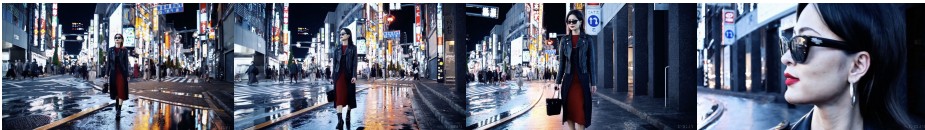

**RACCooN Generated Paragraphs**

The video shows a woman walking down a busy city street at night. She is wearing a red dress and carrying a handbag. She stops to look at a building and then walks away. The woman is also seen walking down a rain-soaked street and looking at a building. She then walks away again. The video also shows a man walking down the street and looking at a building. The woman is also seen walking down a street and looking at a building. The video ends with a woman walking down a street and looking at a building. Overall, the video captures the busy and bustling atmosphere of a city at night. The woman's red dress and handbag stand out as she walks down the street, and her attention to the buildings she looks at adds depth to the scene. The man walking down the street also adds to the overall sense of a busy urban environment.

Object Catalog:
1. Woman: A stylish woman wearing a black leather jacket, a long red dress, and black boots. She carries a black purse and has sunglasses on.
2. Traffic lights: There are traffic lights visible in the background, indicating that the street is a busy intersection.
3. Building: A large, multi-story structure with a prominent presence in the background. It might be a commercial or residential building.
4. Handbag: A small, handheld accessory carried by the woman. It could be used for holding personal belongings or as a fashionable accessory.

Figure 4: **Qualitative V2P example of our RACCooN** on Sora video.

(**DAVIS** (Pont-Tuset et al., 2017), **VPLM**), and conditional video generation (**ActivityNet** (Krishna et al., 2017), **YouCook2** (Zhou et al., 2018a), **UCF101** (Soomro et al., 2012)).

**Metrics**: For each task, we evaluate our approach with various metrics. (**1**) **Video Caption:** following previous works (Yang et al., 2023; Zhou et al., 2018b), we conduct a comprehensive human evaluation and adopt general metrics for our long video descriptions, including SPICE (Anderson et al., 2016), BLEU-4 (Vedantam et al., 2015), and CIDEr (Vedantam et al., 2015). (**2**) **Video Object Layout Planning:** following the prior work (Lin et al., 2023b), we evaluate the framework for object layout planning by bounding box IoU, FVD (Unterthiner et al., 2019), and CLIP-score (Radford et al., 2021). (**3**) **Text-based Video Content Editing:** following prior works (Geyer et al., 2023; Ceylan et al., 2023; Yang et al., 2024), we evaluate the framework by CLIP-Text, CLIP-Frame, Qedit (Yang et al., 2024), and SSIM (Hore & Ziou, 2010). (**4**) **Conditional Video Generation:** we measure FVD (Unterthiner et al., 2019), CLIP-Score (Radford et al., 2021), and SSIM (Hore & Ziou, 2010).

**Implementation Details**: In V2P generation, we set $k = [20, 25]$ and $v = [5, 6]$ for superpixel clustering. We use CLIP-L/14@336 (Radford et al., 2021) as the image encoder and Vicuna-1.5 (Zheng et al., 2024) as the LLM. Our P2V model is started from StableDiffusion-2.0-Inpainting (Rombach et al., 2022). We split the VPLM datasets into train and test sets, with the test set containing 50 unique video-paragraph pairs (for V2P) and 180 mask-object-description triples (for P2V). We manually annotate the editing prompts for the object-changing subtask. We quantitatively compare RACCooN and other baselines on the VLPM test set. To focus on generation results rather than grounding ability, we apply the same ground truth masks and captions to all methods for P2V evaluation. See the Appendix for more details on datasets, metrics, implementations, ablations, and qualitative analysis.

### 4.1 VIDEO-TO-PARAGRAPH GENERATION

**Video-Paragraph Alignment.** We conducted a quantitative evaluation of our proposed RACCooN framework's video-to-paragraph generation capabilities, comparing it against strong baselines with a focus on object-centric captioning and object layout planning. The results, summarized in Tab. 1, show that open-source video-LLMs (e.g., PG-VL, Video-Chat) which have smaller LLMs ($< 13B$

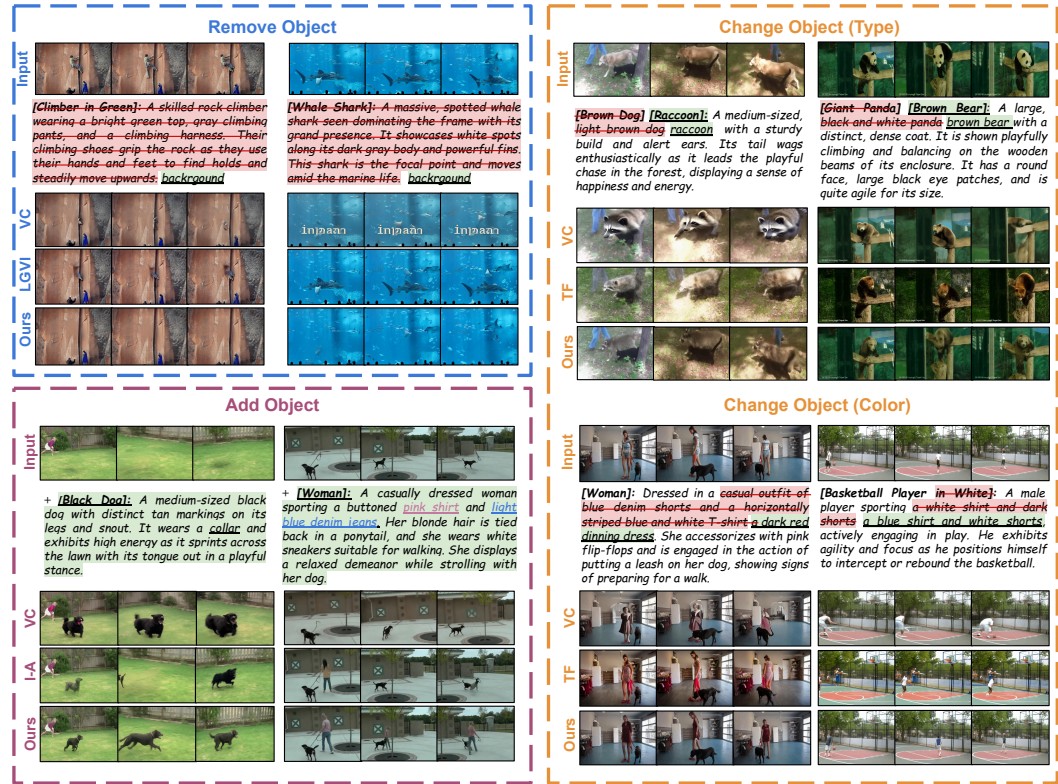

Figure 5: **Qualitative Comparison between RACCooN and other baselines.** Baseline names are abbreviated: **VC**: VideoComposer, **I-A**: Inpainting Anything, **TF**: TokenFlow. We underlined visual details in our caption. More visualizations are in Appendix.

Table 3: **Results of Video Content Editing on three sub-tasks** on VPLM test. We gray out models that conduct the DDIM inversion process and have a different focus on our inpainting-based model.

| Model | Change Object | | | Remove Object | | | Add Object | | |
|---|---|---|---|---|---|---|---|---|---|
| | CLIP-T↑ | CLIP-F↑ | Qedit↑ | FVD↓ | SSIM↑ | PSNR↑ | FVD↓ | SSIM↑ | PSNR↑ |
| *Inversion-based Models* | | | | | | | | | |
| LOVECon (Liao & Deng, 2023) | 29.36 | 94.77 | 1.29 | 1319.51 | 60.40 | 17.78 | 1433.12 | 58.51 | 17.35 |
| FateZero (Qi et al., 2023) | 25.18 | 94.47 | 1.01 | 1037.05 | 47.35 | 15.16 | 1474.80 | 47.65 | 15.45 |
| TokenFlow (Geyer et al., 2023) | 29.25 | 96.23 | 1.31 | 1317.29 | 47.06 | 15.83 | 1373.20 | 49.95 | 15.95 |
| *Inpainting-Based Models* | | | | | | | | | |
| Inpaint Anything (Yu et al., 2023) | 24.86 | 92.01 | 1.01 | 383.81 | 82.33 | 27.69 | 712.59 | 77.75 | 22.41 |
| LGVI (Wu et al., 2024) | 23.82 | **95.33** | 1.04 | 915.24 | 56.16 | 19.14 | 1445.43 | 47.93 | 16.09 |
| VideoComposer (Wang et al., 2024b) | 27.61 | 94.18 | **1.25** | 827.04 | 47.34 | 17.55 | 1151.90 | 48.01 | 15.76 |
| PGVL + SD-v2.0-inpainting | 24.01 | 90.11 | 1.01 | 282.31 | 82.33 | 27.69 | 1579.65 | 43.21 | 15.76 |
| **RACCooN** | **27.85** | 94.78 | 1.15 | **162.03** | **84.38** | **30.34** | **415.82** | **77.81** | **23.38** |

parameters), struggle with object-centric captioning and usually fail to generate layout planning. This is primarily due to their lack of instructional fine-tuning and insufficient video detail modeling without multi-granular pooling. In contrast, our RACCooN framework demonstrates superior performance in both object-centric captioning and complex object layout planning, benefiting from the instructional tuning on our VPLM dataset. Additionally, our method achieves competitive performance with proprietary MLLMs (e.g., Gemini 1.5 Pro, GPT-4o) in key object captioning and layout planning, demonstrating its superior instruction following and generation quality. **Human Evaluation & Qualitative Examples.** We conducted a human evaluation to compare our auto-generated captions with those from a strong baseline and human annotations on ten randomly selected YouCook2 videos (each three to five minutes long with multiple scenes and complex viewpoints). Five evaluators rated these based on Logic Fluency, Language Fluency, Video Summary, and Video Details (details in Appendix). The average scores for each criterion and their overall mean are illustrated in Tab. 2. Our method significantly surpassed both the PG-VL-generated and ground truth captions in all metrics,

showing a **4.9%p** and **21.8%p absolute improvement** respectively, and matched the ground truth in capturing *Video Details* with a **9.4%p** enhancement over the baseline, highlighting RACCooN's superior capability in capturing video details. We additionally visualize descriptions generated by our RACCooN. We use a well-known generated video from the Sora (openai, 2024) generated demo example. As shown in Fig. 4, it demonstrates our model's robust capability to auto-describe complex video content without human textual input.

## 4.2 INSTRUCTIONAL VIDEO EDITING WITH RACCooN

**Quantitative Evaluation.** As shown in Tab. 3, we quantitatively compare the video editing ability of RACCooN with strong video editing models based on inpainting or DDIM-inversion (Hertz et al., 2022) across three object-centric video content editing subtasks: *object changing*, *removal*, and *adding*. In general, RACCooN outperforms all baselines across 9 metrics. For object changing, RACCooN outperforms the best-performing baseline by 0.8% on CLIP-T, indicating better video-text alignment while maintaining temporal consistency, as demonstrated by comparable CLIP-F and Qedit scores. Note that LGVI is not designed to alter video attributes and tends to preserve video content with marginal change (i.e., identical input and output videos), resulting in improved CLIP-F scores. In the object removal task, RACCooN shows significant improvements over strong baselines (relatively $+57.8\%$ FVD, $+2.5\%$ SSIM, $+9.6\%$ PSNR). Such improvements are maintained in the addition task (relatively $+41.6\%$ FVD, $+4.3\%$ PSNR). Meanwhile, some DDIM inversion-based models (e.g., TokenFlow (Geyer et al., 2023)) work well for specific tasks (change objects), but do not handle other types of editing. In contrast, our method is an all-rounder player.

We further emphasize that **a simple combination of existing models cannot achieve an effective framework**, leading to inferior instruction-following and editing abilities. This is evident in the degraded performance of open-source Video-LLM baselines (Tabs. 1 and 2) and other video editing models (Tab. 3). To address these limitations, we made unique novelty in technical and dataset contributions and achieved significant improvements in both video understanding and editing. This is evident in the comparison of our RACCooN with a multi-agent baseline combining a powerful open-source video reasoning framework and video diffusion models (PG-Video-LLaMA + StableDiffusion 2.0-inpainting) in Tab. 3. RACCooN outperforms the *PGVL + SD 2.0-inpainting* by a significant margin across all metrics and editing tasks, highlighting the effectiveness of our proposed framework.

**Visualization.** In Fig. 5, we compare videos generated by RACCooN with several SoTA baselines across three video content editing tasks. For object removal, RACCooN demonstrates superior results, naturally and smoothly inpainting the background, whereas VideoComposer generates unexpected content and LGVI fails to accurately remove objects across frames. For object addition, compared to Inpainting-Anything and VideoComposer, which often miss objects or produce distorted generations, RACCooN generates objects with more fluent and natural motion, accurately reflecting caption details (e.g., the *collar* of the dog, the *pink shirt*, and *blue jeans* for the woman). For changing objects, our method outperforms inpainting-based VideoComposer and inversion-based TokenFlow. RACCooN accurately re-paints objects to achieve object editing for color (*white→blue*) and type (*dog→RACCooN*), while others struggle to meet requirements.

**Ablation Studies.** As shown in Tab. 4, we further validate the effectiveness of components by replacing detailed descriptions with short captions, and oracle masks/planning boxes with model-generated ones. In adding objects, our detailed object descriptions can benefit generation by providing accurate details, leading to improved quantitative results (relatively $+14.4\%$ FVD). We further replace GT boxes with boxes predicted by RACCooN, and still show superior performance over other baseline methods with oracle boxes in Tab. 3. It demonstrates that our V2P stage can thus automatically generate planning from a given video to eliminate users' labor. Next, in object removal and changing, we replace the oracle masks with grounding (Liu et al., 2023d) and tracking (Cheng et al., 2023) tools generated mask,

Table 4: **Ablation on video object changing, removing, and adding** with different inputs.

| Settings | FVD↓ | SSIM↑ | PSNR↑ |
|---|---|---|---|
| *add object* | | | |
| RACCooN | 415.80 | 77.81 | 23.38 |
| w/o detail caption | 476.01 | 76.80 | 23.14 |
| w/o oracle planning | 969.95 | 76.65 | 21.21 |
| *remove object* | | | |
| RACCooN | 162.03 | 84.38 | 30.34 |
| w/o oracle mask | 398.01 | 81.60 | 27.15 |
| Setting | CLIP-T | CLIP-F | Qedit |
| *change object* | | | |
| RACCooN | 27.85 | 94.78 | 1.15 |
| w/o oracle mask | 27.23 | 94.33 | 1.14 |

Table 5: **Results of Inversion-based Video Editing** on DAVIS Video. Our generated paragraph can be integrated with different SoTA inversion-based video editing models (e.g. TokenFlow or Fate-Zero). *attr.* and *ins.* indicate *attribute-* and *instance-level* editing.

| Model | CLIP-Text | | | CLIP-Frame | | | SSIM | | |
|---|---|---|---|---|---|---|---|---|---|
| | attr. | ins. | all | attr. | ins. | all | attr. | ins. | all |
| FateZero (Qi et al., 2023) | 28.9 | 27.2 | 28.1 | **95.6** | **95.1** | **95.3** | 71.5 | 70.8 | 71.2 |
| FateZero (Qi et al., 2023) + RACCooN | **31.7** | **30.7** | **31.2** | 95.5 | **95.1** | **95.3** | **72.1** | **72.3** | **72.2** |
| TokenFlow (Geyer et al., 2023) | 31.4 | 29.8 | 30.6 | 94.6 | 94.1 | 94.3 | 57.0 | 56.0 | 56.5 |
| TokenFlow (Geyer et al., 2023) + RACCooN | **32.6** | **31.6** | **32.1** | **94.7** | **94.3** | **94.5** | **58.0** | **57.3** | **57.6** |

Table 6: **Results of Conditional Video Generation** on three datasets. RACCooN framework can be integrated with different video generation models (e.g. VideoCrafter or DynamiCrafter.

| Model | ActivityNet | | | YouCook2 | | | UCF101 | | |
|---|---|---|---|---|---|---|---|---|---|
| | FVD↓ | CLIP↑ | SSIM↑ | FVD↓ | CLIP↑ | SSIM↑ | FVD↓ | CLIP↑ | SSIM↑ |
| VideoCrafter (Chen et al., 2023) | 3743.62 | 9.58 | 22.26 | 4731.22 | **9.89** | 20.58 | 3556.06 | **9.81** | 18.32 |
| VideoCrafter (Chen et al., 2023) + RACCooN | **2357.41** | **10.53** | **24.02** | **3046.82** | 9.47 | **23.89** | **2208.90** | 9.60 | **22.32** |
| DynamiCrafter (Xing et al., 2023) | 1632.30 | 10.65 | 32.46 | 2059.93 | **11.95** | 37.22 | 1588.57 | **11.98** | 38.81 |
| DynamiCrafter (Xing et al., 2023) + RACCooN | **1536.63** | **10.69** | **32.86** | **1904.08** | 10.03 | **38.78** | **1573.27** | 9.76 | **39.83** |

it shows marginally decrement for changing objects, and our framework still shows strong results over other baselines in Tab. 3 with oracle masks. It suggests that RACCooN is effective and robust to handle diverse editing skills in a non-orcale setting (See the Appendix).

### 4.3 ENHANCING INVERSION-BASED VIDEO EDITING & GENERATION WITH RACCOON

We further validate that detailed paragraphs can benefit video generative tasks (e.g., inversion-based video editing and video generation). Our RACCooN captions can be integrated with off-shelf SoTA video editing and generation models to enhance them. See Appendix for more details.

**Inversion-based Video Editing.** Our RACCooN framework can significantly enhance video editing tasks. We integrated it into two SoTA methods, TokenFlow (Geyer et al., 2023) and FateZero (Qi et al., 2023), and compared their performance with different text inputs. The baseline used human-written short captions, while *baseline*+RACCooN used generated detailed captions. As shown in Tab. 5, integrating RACCooN significantly improved performance. FateZero+RACCooN achieved an 11.0% and 1.4% relative increase in CLIP-Text and SSIM scores, respectively. TokenFlow+RACCooN saw increases of 4.9% and 1.9%. These results indicate that detailed captions enhance text-to-video alignment and localized editing, while preserving temporal consistency and quality.

**Conditional Video Generation.** We also integrated RACCooN with various SoTA video generation methods, such as VideoCrafter (Chen et al., 2023) and DynamiCrafter (Xing et al., 2023). VideoCrafter was tested in a Text-to-Video setting, while DynamiCrafter was in an Image-Text-to-Video setting. We evaluated our framework on ActivityNet, YouCook2, and UCF101 datasets. As listed in Tab. 6, VideoCrafter's generated videos often failed to align with the GT captions, especially in complex videos. In contrast, our framework improved video quality and alignment with auto-generated detailed paragraphs, achieving substantial improvements of 36.9% in FVD and 15.3% in SSIM. We noted consistent improvements in FVD, CLIP, and SSIM scores with RACCooN. It underscores our effectiveness in augmenting video generation models with automated descriptions.

## 5 CONCLUSION

Our proposed RACCooN framework newly introduces an auto-descriptive video-to-paragraph-to-video generative framework. RACCooN automatically generates video descriptions by leveraging a multi-granular spatiotemporal pooling strategy, enhancing the model's ability to recognize detailed, localized video information. RACCooN then uses these enriched descriptions to edit and generate video content, offering users the flexibility to modify content through textual updates, thus eliminating the need for detailed video annotations. These video editing and generation abilities of RACCooN framework highlight notable effectiveness and enable a broader range of users to engage in video creation and editing tasks without the faithfully written textual prompts.

ETHICS STATEMENT

The performance of our proposed framework in paragraph generation, video generation, and editing is influenced by the employed pre-trained backbones, including an LLM (Touvron et al., 2023). LLM-empowered video description and photorealistic video creation/editing inherit biases from their training data, leading to several potentially negative impacts, including societal stereotypes, biased interpretation of actions, and privacy concerns. To mitigate these potential negative impacts, it is essential to carefully develop and implement generative and video description models, such as considering diversifying training datasets, implementing fairness and bias evaluation metrics, and engaging communities to understand and address their concerns.

REPRODUCIBILITY STATEMENT

This paper fully discloses all the information needed to reproduce the main experimental results of the paper to the extent that it affects the main claims and/or conclusions. To maximize reproducibility, we have included our code in the supplementary material. Also, we report all of our hyperparameter settings and model details in the Appendix.

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

# APPENDIX

In this appendix, we present the following:

- More details about VPLM dataset collection (Sec. A.1), experimental setups (Sec. A.2), more implementation details (Sec. A.3).
- Limitations and Negative Societal Impact of RACCooN (Sec. B).
- Additional analysis including ablations (Sec. C.2, Sec. C.1).
- Additional qualitative examples with RACCooN on video content editing (Sec. C.3).

# A EXPERIMENTAL SETUP

**GPT4 Prompt**

Please carefully analyze the provided video. Please carefully analyze the provided video. The video is presented as frame sequences, and they are placed in a top-left to bottom-right order as the temporal order (the top-left is the start frame, the bottom-right is the end frame, and there are frame indexes 1,2,3,4,5,6 on each frame).

Please focus on two tasks (1) providing a short and well-organized holistic video description including main objects, actions, and events; (2) identifying and cataloging (up to five) significant objects within this video.

Ensure that your analysis adheres to the structure outlined below, prioritizing objects that are clear, discernible, and integral to the overall context of the video. Your descriptions should be rich in detail, capturing key attributes such as color, shape, actions, styles, movements, and any distinct features. Strive for precision and succinctness to aid in comprehension and ease of reference.

Format for Video Description (Ensure to avoid repetitive descriptions and avoid frame-by-frame descriptions.):

Video Description: Drone view of waves crashing against the rugged cliffs along Big Sur's garay point beach. The crashing blue waters create white-tipped waves, while the golden light of the setting sun illuminates the rocky shore. A small island with a lighthouse sits in the distance, and green shrubbery covers the cliff's edge. The steep drop from the road down to the beach is a dramatic feat, with the cliff's edges jutting out over the sea. This is a view that captures the raw beauty of the coast and the rugged landscape of the Pacific Coast Highway.

Example Format for Listing Identified Object Names and their Descriptions:

1. Woman: A stylish woman. She wears a black leather jacket, a long red dress, and black boots, and carries a black purse. She wears sunglasses and red lipstick. She walks confidently and casually.
    ... (Continue if there are other significant objects)

Hints for Objects in the Video:
1. Panda
….
Number each object uniquely, ensuring your descriptions are both concise and thorough, providing clear insight into each object's appearance and its significance within the video. Highlight any relationships between objects if they are part of a larger, significant item or scene.
Your descriptions should excel in clarity, detail, and specificity, ensuring you can reconstruct the objects in your mind according to your descriptions.
Ensure the output contains objects in the provided hints. Ensure the output length is less than 400 words. Do not provide explanations or notes for the generated captions.

SuperImage as Video

**Video Description**
A caretaker at a wildlife sanctuary is engaging with a giant panda in a lush green bamboo forest. The video depicts a series of frames showing the caretaker in a light grey uniform, approaching and interacting with the panda. He gently assists the panda as it climbs over a log structure designed to mimic the animal's natural habitat. The environment is serene, surrounded by dense greenery and the sounds of nature, focusing on conservation and animal care.
**Identified Object Names and Descriptions**
**1. Giant Panda:** A large, distinctive black and white bear known for its docile temperament and strong association with conservation efforts. Its black ears, eye patches, legs, and a band across its shoulders are contrasted sharply with its white face and body.

**2. Caretaker:** An adult male wearing a light grey uniform jacket with green embroidery symbolizing a panda and some script, possibly identifying the organization. He also wears a dark belt and light-colored pants, showing his official role in animal care and maintenance.

**3. Bamboo Forest:** Dense and lush, the forest comprises tall, green bamboo stalks that provide a naturalistic and enriching habitat for the panda. The area exudes tranquility and is pivotal for the ecological theme of the sanctuary.

Figure 6: **Pipeline of our VPLM dataset annotation with GPT4V**. We first convert a video as a superimage and then give some in-context examples to prompt GPT-4V to annotate detailed and well-structured video descriptions.

## A.1 VPLM DATA COLLECTION.

As we mention in Sec. 3.3, to facilitate our model training, we start from open-source video inpainting data (Wu et al., 2024)[3] to build a high-quality dataset that includes the well-structured, detailed caption for both video and each object in the video. Specifically, we leverage GPT-4V [4] to annotate each video. As shown in Fig. 6, we first convert a video to a superimage (Fan et al., 2021) by concatenating uniformly sampled frames, and we also draw frame IDs on each frame as a visual prompt to present temporal order. Then we prompt the GPT-4V by providing a detailed prompt with in-context examples (left of Fig. 6). In this case, we obtained well-structured, detailed captions that contain both holistic video and local objects (bottom right of Fig. 6). To ensure the annotation quality, we sampled 1 annotated video from each 100 batches and then did a human cross-check, and refined the batch annotations according to the sampled example. Through this pipeline, we obtained 7.2K high-quality quality well-structured, detailed video descriptions with an average of 238.0 words for each video.

Next, to obtain paired object-mask-description triplets for video inpainting model training, we build an automatic detailed object caption and object name matching pipeline using GPT4. As in our base

---

[3]MIT License: https://choosealicense.com/licenses/mit/

[4]version 1106

dataset (Wu et al., 2024), we already have class labels for each object mask, we framed this matching as a multi-choice QA to ask GPT4 which object caption can in Fig. 6 matched to the given object classes. We further filtered out the triplets with too small masks ($< 1\%$ mask areas.) In this case, we obtained 5.5K object-mask-description triplets with an average of 37.2 words for each object to support our video diffusion model training.

## A.2 BENCHMARKS AND DATASETS DETAILS

As mentioned in the main paper (**??**), we evaluate our proposed RACCooN on various tasks. For video-to-paragraph generation, we test our model on the standard video caption dataset YouCook2 (Zhou et al., 2018a) (validation set) as well as our VPLM dataset. We next test video content editing with three subtasks on our VPLM dataset. Regarding the experiments of incorporating RACCooN with other conditional video generation models, we test RACCooN on diverse videos from ActivityNet, YouCook2, and UCF101. We uniformly selected 100 videos from those 3 datasets to build the test bed. For the experiments of incorporating RACCooN with other video editing models, we follow the previous work (Geyer et al., 2023), and select 30 unique videos from the DAVIS dataset. For each video, we annotate two different types of editing, attribute editing and instance editing. It leads to 60 text-video pairs in our video editing evaluation. We choose object captions that contain the same keywords for editing in human captions to represent the model-generated caption.

## A.3 IMPLEMENTATION DETAILS

**Metrics**: We provide more details about our metrics. CLIP-Text measures the similarity between the edit prompt and the embedding of each frame in the edited video. CLIP-Frame computes the average CLIP similarity between the image embeddings of consecutive frames to measure the temporal coherency. SSIM measures the structural similarity between the original and edited video for evaluating localized editing. $Q_{edit} = CLIP - T/Wrap - Err$, it is a comprehensive score for video editing quality, where $Warp - Err$ calculates the pixel-level difference by warping the edited video frames according to the estimated optical flow of the source video, extracted by FlowNet2.0 (Ilg et al., 2017). For layout planning, we compute the FVD and CLIP-Image similarity between the ground truth and the predicted bounding box.

**More Details about Multi-granular Spatiotemporal Pooling.** As mentioned Sec. 3.1, we proposed a novel Multi-granular Spatiotemporal Pooling (MGS Pooling) to address the lack of complex spatial-temporal modeling in video. We further provide a more intuitive visualization for our proposed MGS Pooling in Fig. 3. We first adopt a lightweight 165 superpixel predictor that generates superpixels across video frames, then use the overlapping k-means clustering for the obtained video superpixels. In this case, we gather informative cues about various objects and actions for LLM.

**Human Evaluation on Video-to-Paragraph Generation.** We conduct a human evaluation on nine randomly selected videos from the YouCook2 dataset. Videos are three- to five-minute-long and contain multiple successive scenes with complex viewpoints. We provide these videos to four different annotators with ground truth captions, descriptions generated by PG-Video-LLAVA, and our method, RACCooN, where the captions/descriptions for each video are randomly shuffled. We leverage four distinct human evaluation metrics: Logic Fluency (Logic), Language Fluency (Language), Video Summary (Summary) and Video Details (Details). To avoid a misinterpretation of methods' capabilities due to relative evaluation, we instruct the annotators to independently rate the quality of each set of captions based on these four different criteria, by giving a score from 1 to 5 (i.e., choice: [1, 2, 3, 4, 5]).

**Off-shelf Video Editing Models.** We utilize TokenFlow (Geyer et al., 2023) and FateZero (Qi et al., 2023) as our video editing tools. TokenFlow generates a high-quality video corresponding to the target text, while preserving the spatial layout and motion of the input video. For SSIM computation, we compute SSIM for region-of-no-interest since we want to keep those regions unchanged. we first mask out regions of interest with the ground truth mask provided by the DAVIS dataset, the we compute SSIM on masked images and conduct mean pooling as the video-level metrics.

**Off-shelf Conditional Video Generation Models.** We leverage both VideoCrafter (Chen et al., 2023) and DynamiCrafter (Xing et al., 2023) as our video generation backbone. VideoCrafter is one of the SoTA video generation models that can handle different input conditions (image, text).

DynamiCrafter is based on the open-source VideoCrafter and T2I Latent Diffusion model (Rombach et al., 2022), and was trained on WebVid10M (Bain et al., 2021), it provides better dynamic and stronger coherence. We adopt VideoCrafter-512 and DynamiCrafter-512 variants. For each video, we use CLIP similarity to retrieve multiple keyframes corresponding to each caption. Those keyframes result in multiple generated video clips via the video generation model. For FVD computation, we conduct mean pooling over those clips to represent a video. We use $k = 25$ and $v = 6$ for generated captions in all experiments. The experiments are conducted on the $4 \times 48$GB A6000 GPUs machine.

## B  LIMITATIONS AND BROADER IMPACT

Our proposed RACCooN framework has shown a remarkable ability to interpret input videos, producing well-structured and detailed descriptions that outperform strong video captioning baselines and even ground truths. However, it has the potential to produce inaccuracies or hallucination (Liu et al., 2023a; Wang et al., 2024a; Zhou et al., 2024; Ma et al., 2023) in the generated text outputs. In addition, the performance of our proposed framework in paragraph generation, video generation, and editing is influenced by the employed pre-trained backbones, including an LLM (Touvron et al., 2023), base Inpainting Model (Rombach et al., 2022), Video Diffusion Model (Xing et al., 2023), and Video Editor (Geyer et al., 2023). However, our key contributions are independent of these backbones, and we emphasize that RACCooN's capabilities can be further enhanced with future advancements in these generative model backbones.

LLM-empowered video description and photorealistic video creation/editing inherit biases from their training data, leading to several broader impacts, including societal stereotypes, biased interpretation of actions, and privacy concerns. To mitigate these broader impacts, it is essential to carefully develop and implement generative and video description models, such as considering diversifying training datasets, implementing fairness and bias evaluation metrics, and engaging communities to understand and address their concerns.

## C  ADDITIONAL ANALYSIS

### C.1  ABLATION STUDY

**The effect of $k$ and $v$** As shown in Tab. 7, we did initialized hyperparameter probing experiments on ActivityNet-Cap and YouCook2 datasets. we observe that all variants of our approach with varying $k$ and $v$ generally achieve improved performance compared to baselines in terms of multiple video captioning metrics: *SPICE*, *BLEU-4*, *METEOR*, and *ROUGE*. This result demonstrates the efficacy of our multi-granular spatiotemporal pooling approach with a fine to coarse search of video contexts based on superpixels. In addition, we observe that RACCooN framework shows a small gap between variants in each dataset, highlighting the robustness of our approach to the hyperparameter setups and datasets.

**The effect of Superpixel Overlap** We introduce overlapping k-means clustering to aggregate video superpixels, capturing a variety of visual contexts while allowing for partial spatiotemporal overlap. To investigate the effect of our suggested overlapping approach, we also evaluate the variant of our framework without overlap (i.e., $v = 1$) on video-to-paragraph generation tasks in Tab. 7. As shown, our approach with overlap (i.e., $v > 1$) surpasses the non-overlapping version of RACCooN across various scales of visual contexts $k$, as indicated by the video captioning metrics we evaluated. This emphasizes the advantage of permitting overlap in understanding video contexts, which enhances the input video's comprehension by allowing for diverse and fluent interpretations of local visual regions with surroundings associated at the same time.

For simplicity, we use $k = 25$ and $v = 6$ for all experiments on conditional video generation and video editing tasks, demonstrating the robustness of RACCooN for hyperparameters.

### C.2  COMPARISON WITH PRE-TRAINED GROUNDING MODELS

We further investigate the applicability of recent powerful pre-trained visual grounding models (Kirillov et al., 2023; Cheng et al., 2023; Ren et al., 2024). Segment-Anything (Kirillov et al., 2023)

Table 7: **Ablation of RACCooN** for Video-to-Paragraph Generation on ActivityNet and YouCook2. Metrics are abbreviated: **M**: *METEOR*, **B**: *BLEU-4*, **S**: *SPICE*, **R**: *ROUGE*. $v = 1$ indicates the version without superpixel overlap. We highlight the hyperparameter setup used in the main experiment.

| Models | $k$ | $v$ | ActivityNet | | | | YouCook2 | | | |
|---|---|---|---|---|---|---|---|---|---|---|
| | | | S | B | M | R | S | B | M | R |
| PDVC (Wang et al., 2021) | - | - | - | 2.6 | 10.5 | - | - | 0.8 | 4.7 | - |
| Vid2Seq (Yang et al., 2023) | - | - | 5.4 | - | 7.1 | - | 4.0 | - | 4.6 | - |
| ZeroTA (Jo et al., 2023) | - | - | 2.6 | - | 2.7 | - | 1.6 | - | 2.1 | - |
| PG-VL (Munasinghe et al., 2023) | - | - | 13.6 | 13.9 | 14.2 | 18.1 | 6.2 | 16.5 | 8.6 | 15.8 |
| RACCooN (Ours) | 20 | 1 | 13.5 | 13.9 | 14.2 | 18.1 | 6.3 | 16.9 | 8.7 | 15.9 |
| | | 2 | 13.7 | 14.6 | 14.4 | 18.2 | 6.4 | 17.5 | 8.7 | 16.1 |
| | | 4 | 13.6 | 14.3 | 14.3 | 18.2 | 6.6 | 16.2 | 8.8 | 16.0 |
| | | 5 | 13.8 | 15.0 | 14.5 | 18.4 | 6.4 | 17.9 | 8.8 | **16.2** |
| | 25 | 1 | 13.6 | 14.1 | 14.3 | 18.0 | 6.1 | 16.9 | 8.6 | 15.9 |
| | | 2 | **13.8** | 14.4 | 14.3 | 18.3 | 6.4 | 16.3 | **9.0** | 16.0 |
| | | 4 | 13.6 | 14.3 | 14.3 | 18.2 | 6.6 | 17.1 | 8.9 | 16.1 |
| | | 6 | 13.7 | 14.5 | 14.4 | 18.2 | 6.9 | 18.0 | 9.0 | 16.1 |
| | | 10 | - | - | - | - | 6.4 | 16.5 | 8.7 | 16.1 |
| | 30 | 1 | 13.6 | 14.1 | 14.3 | 18.0 | 6.3 | 16.5 | 8.8 | 16.0 |
| | | 2 | 13.7 | 14.2 | 14.2 | 18.1 | 6.6 | 17.1 | **9.0** | **16.2** |
| | | 4 | 13.5 | 14.5 | 14.3 | 18.2 | 6.6 | **18.1** | 8.8 | 16.1 |
| | | 6 | 13.6 | 14.4 | 14.4 | 18.2 | 6.4 | 17.2 | 8.8 | **16.2** |

Table 8: **RACCooN variants with different grounding methods** for Video-to-Paragraph Generation on YouCook2.

| Method | Localization | Clustering | SPICE | BLEU-4 | METEOR | ROUGE |
|---|---|---|---|---|---|---|
| PG-VL (Munasinghe et al., 2023) | - | - | 6.2 | 16.5 | 8.6 | 15.8 |
| Ours | SAM (Kirillov et al., 2023) | - | 6.4 | 16.9 | 8.7 | 15.9 |
| | Grounded SAM (Ren et al., 2024) | - | 6.5 | 16.5 | 8.7 | **16.1** |
| | SAM-Track (Cheng et al., 2023) | k-means | 6.2 | 16.5 | 8.8 | **16.1** |
| | SAM-Track (Cheng et al., 2023) | overlapping k-means | 6.5 | 17.4 | **9.0** | **16.1** |
| | Superpixel | overlapping k-means | **6.9** | **18.0** | **9.0** | **16.1** |

and Grounded SAM (Ren et al., 2024) are strong open-ended object segmentation models for images, and we directly compute our localized granular tokens based on their segmentation masks. We select 25 segmentation masks in total, from uniformly sampled frames in each video for fair comparison with RACCooN ($k = 25$). As shown in Tab. 8, these variants of RACCooN achieve improved performance against the best-performing baseline, PG-VL, but are often suboptimal since they focus on regional information and cannot contain temporal information of the videos. Unlike these image-based segmentation methods, SAM-Track (Cheng et al., 2023) generates coherent masks of observed objects over successive frames in videos, by adopting multiple additional pre-trained modules, including GroundingDino (Liu et al., 2023d) and AOT (Yang et al., 2021; Yang & Yang, 2022). We adopt SAM-Track to initialize superpixels in videos and conduct overlapping k-means clustering ($k = 25$). Here, we observe that RACCooN with SAM-Track superpixel initialization achieves reasonable performance, and is beneficial for video editing tasks. It enables the model to coherently edit targeted regions in videos with edited keywords.

### C.3 ADDITIONAL VISUALIZATIONS

In this section, we provide more qualitative examples of various tasks, including three types of video content editing, ablation on removing oracle planning and GT masks, enhanced video editing, and conditional video generation.

**Remove, Add, and Change Object the videos.** We provide more qualitative examples in this Appendix across different types of video content editing ( Fig. 7), including removing (Fig. 8, Fig. 9, Fig. 10), adding (Fig. 11, Fig. 12, Fig. 13), and changing/editing (Fig. 14, Fig. 15, Fig. 16). According to the visualization, our RACCooN generally outperforms other strong baselines on all three subtasks. Our RACCooN can reflect the updated text description more accurately, thus aiding in a user-friendly video-generative framework. For example, our method can accurately change the color of the hat (*red→blue*), which is a very small area in the video, while other methods struggle to meet the requirement.

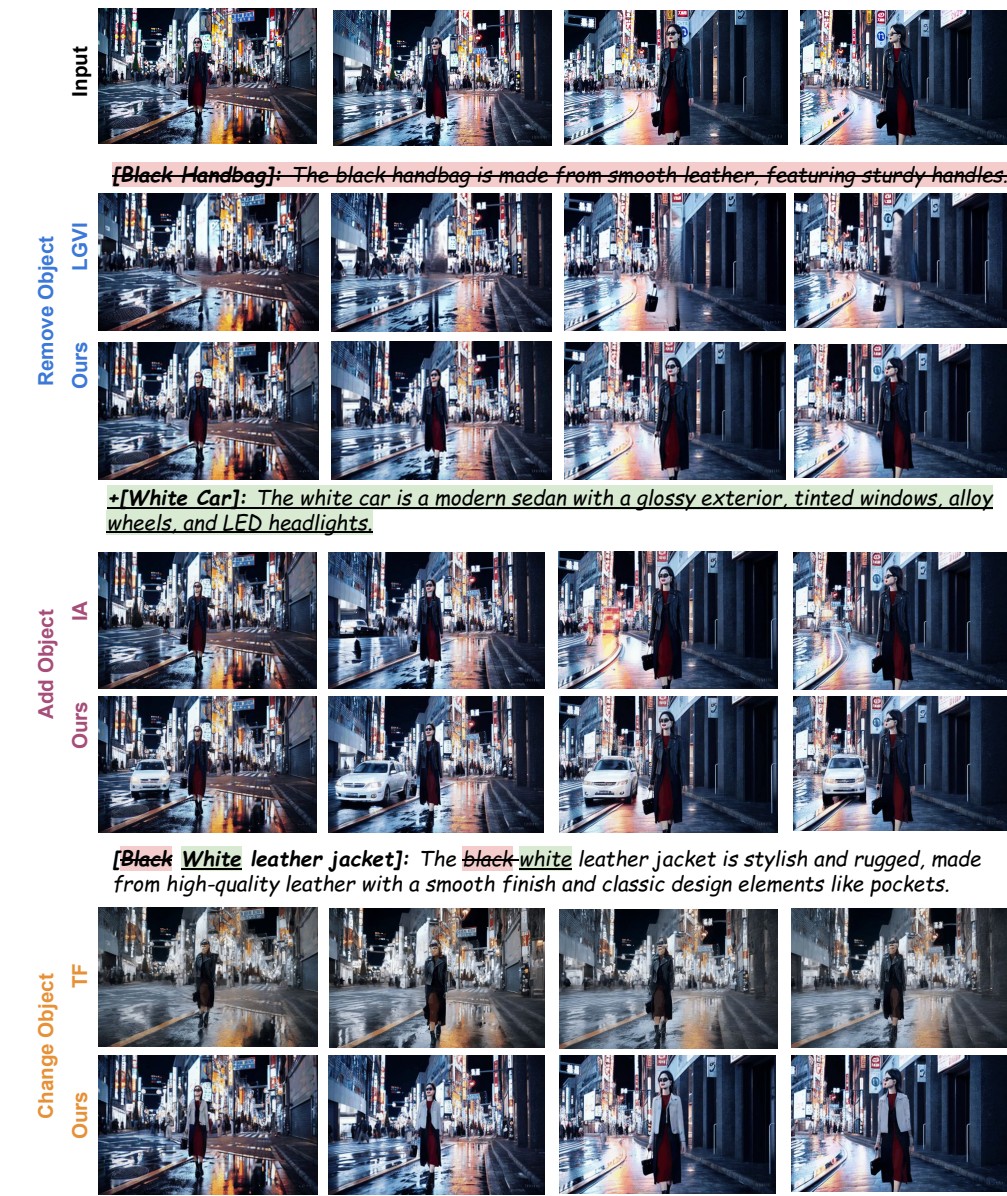

Figure 7: More visualization of diverse editing skills on Sora video and comparison with other methods

**Ablation Study Visualization.** We illustrate extra visualizations for replacing orecle mask with grounding&tracking tools generated ones for video object removal ( Fig. 17 and Fig. 18) and changing ( Fig. 19 and Fig. 20), as well as replacing oracle object boxes with our model-predicted one ( Fig. 21 and Fig. 22). Our framework shows robust results with LLM planning and off-shelf segmentation tools. We further show the failure cases of removing and changing objects mainly come from the missing mask prediction of the video segmentation masks.

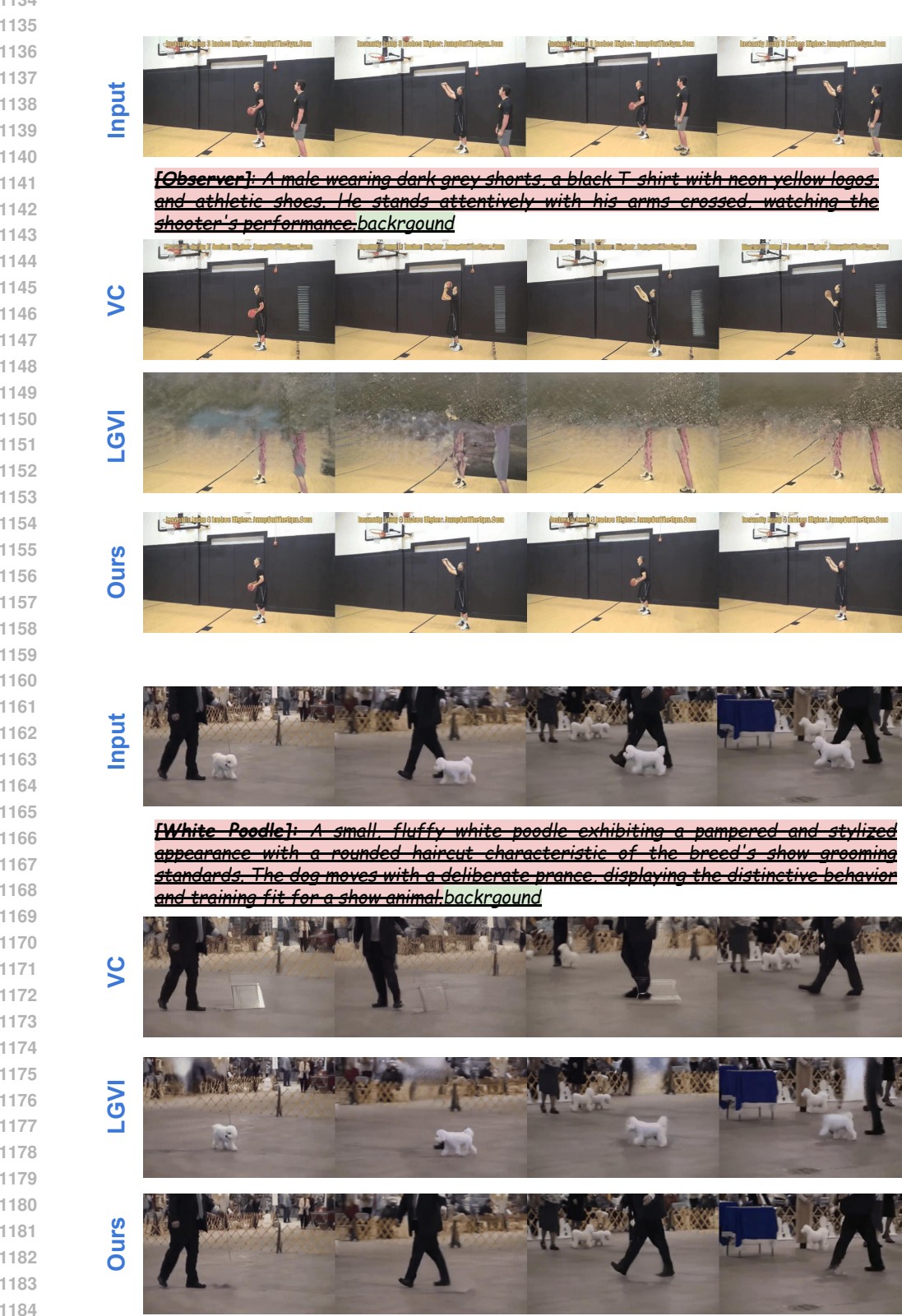

Figure 8: More visualization of **removing** video objects

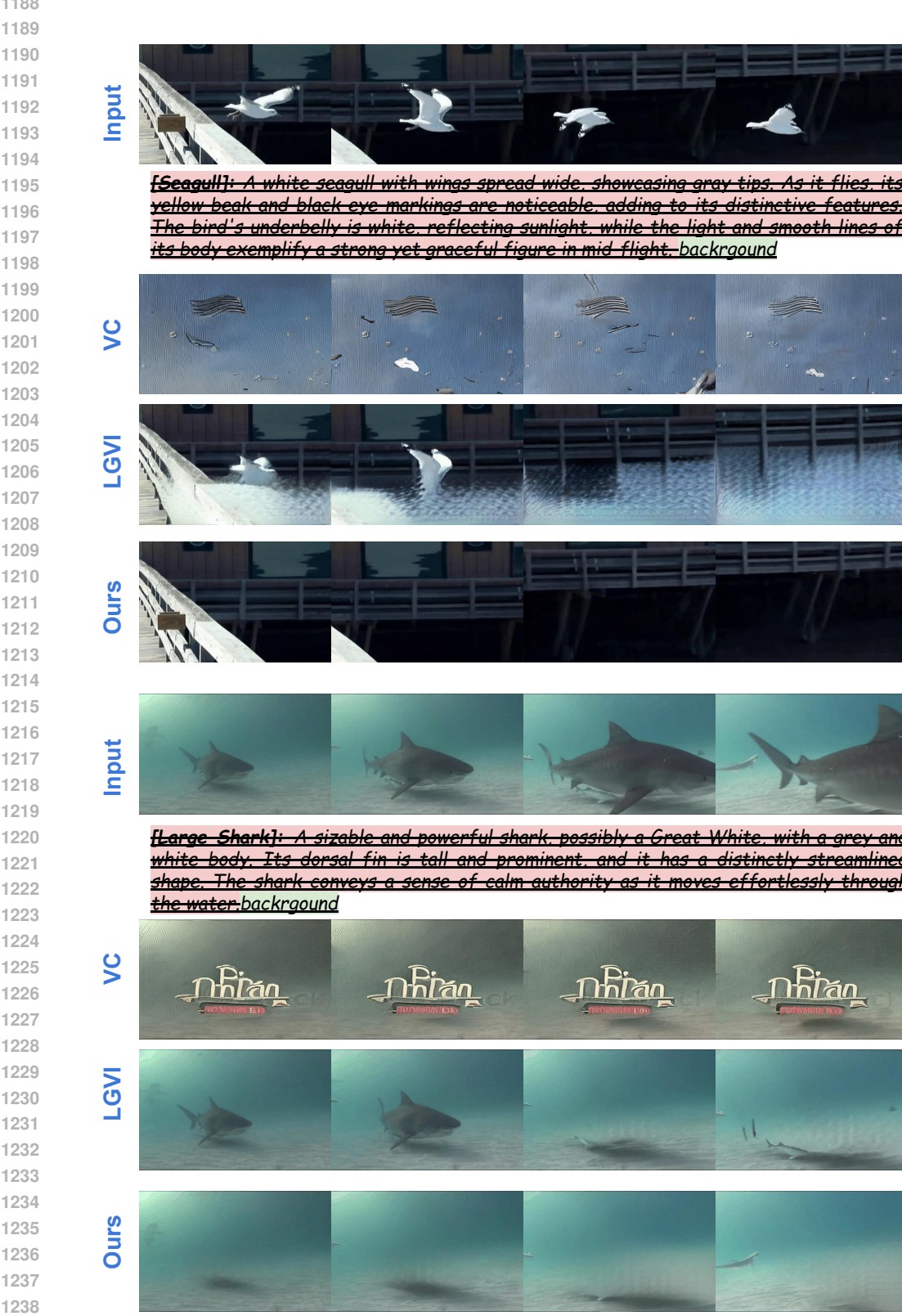

Figure 9: More visualization of **removing** video objects

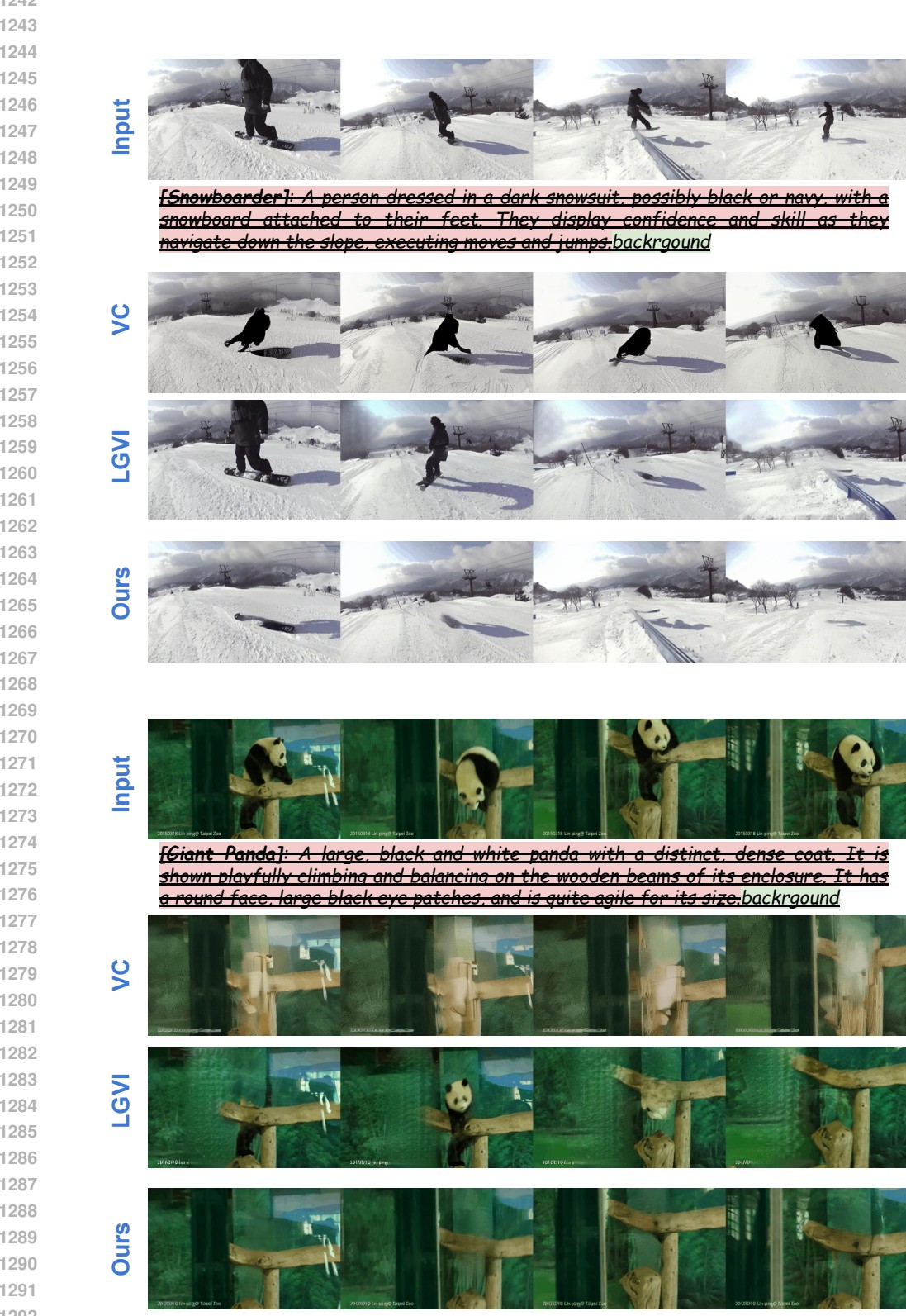

Figure 10: More visualization of **removing** video objects

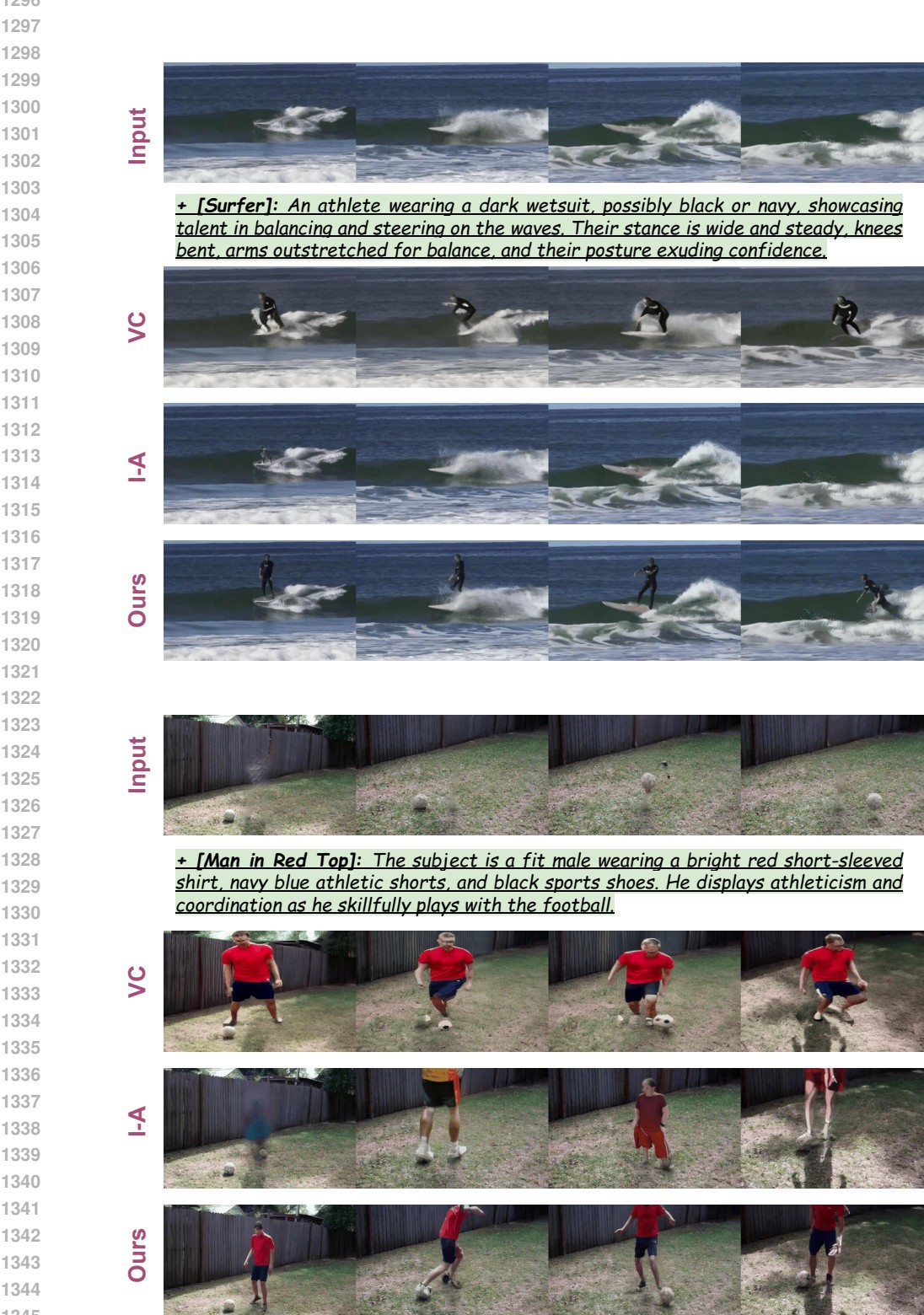

Figure 11: More visualization of **adding** video objects

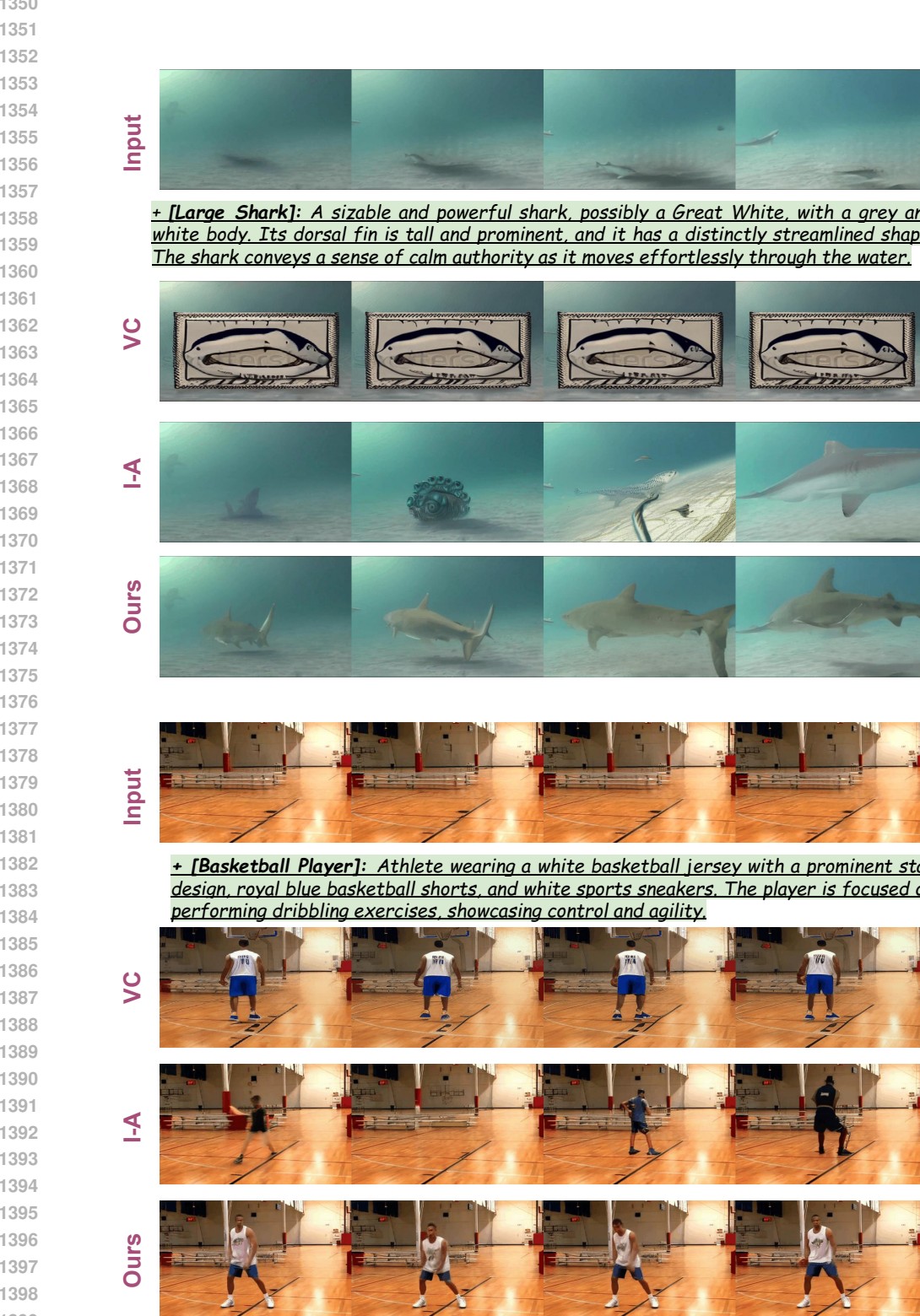

Figure 12: More visualization of **adding** video objects

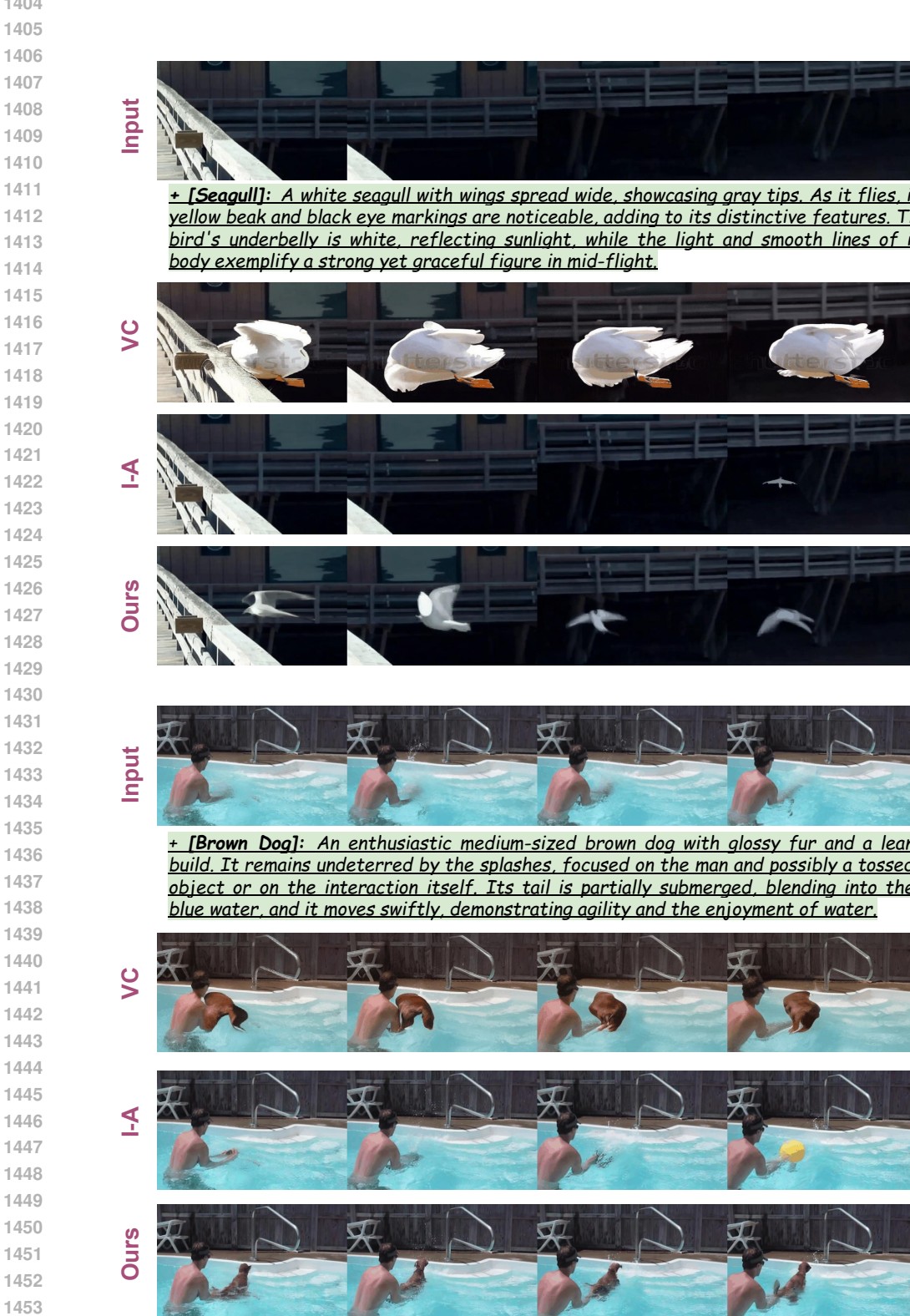

Figure 13: More visualization of **adding** video objects

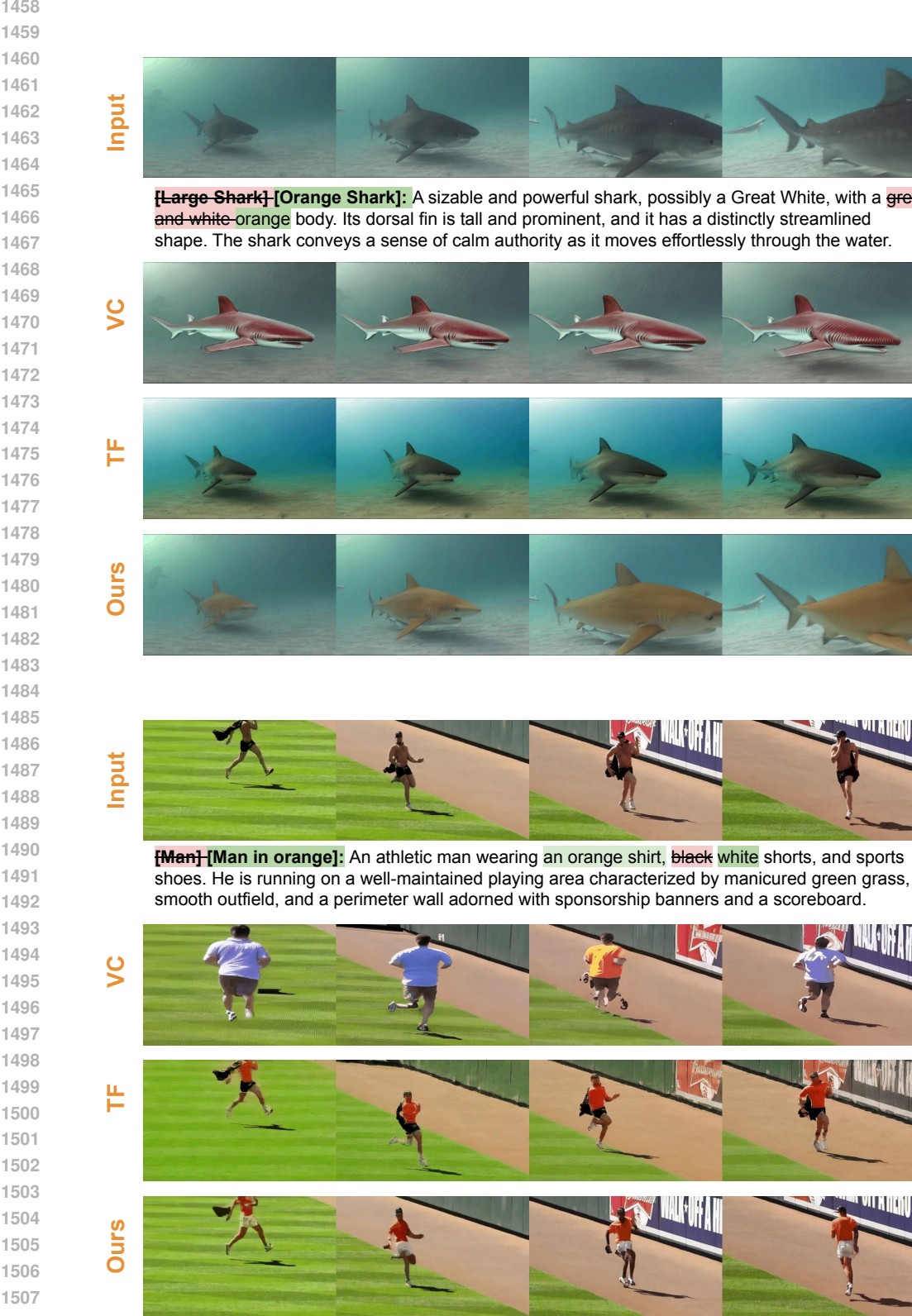

Figure 14: More visualization of **editing** video objects

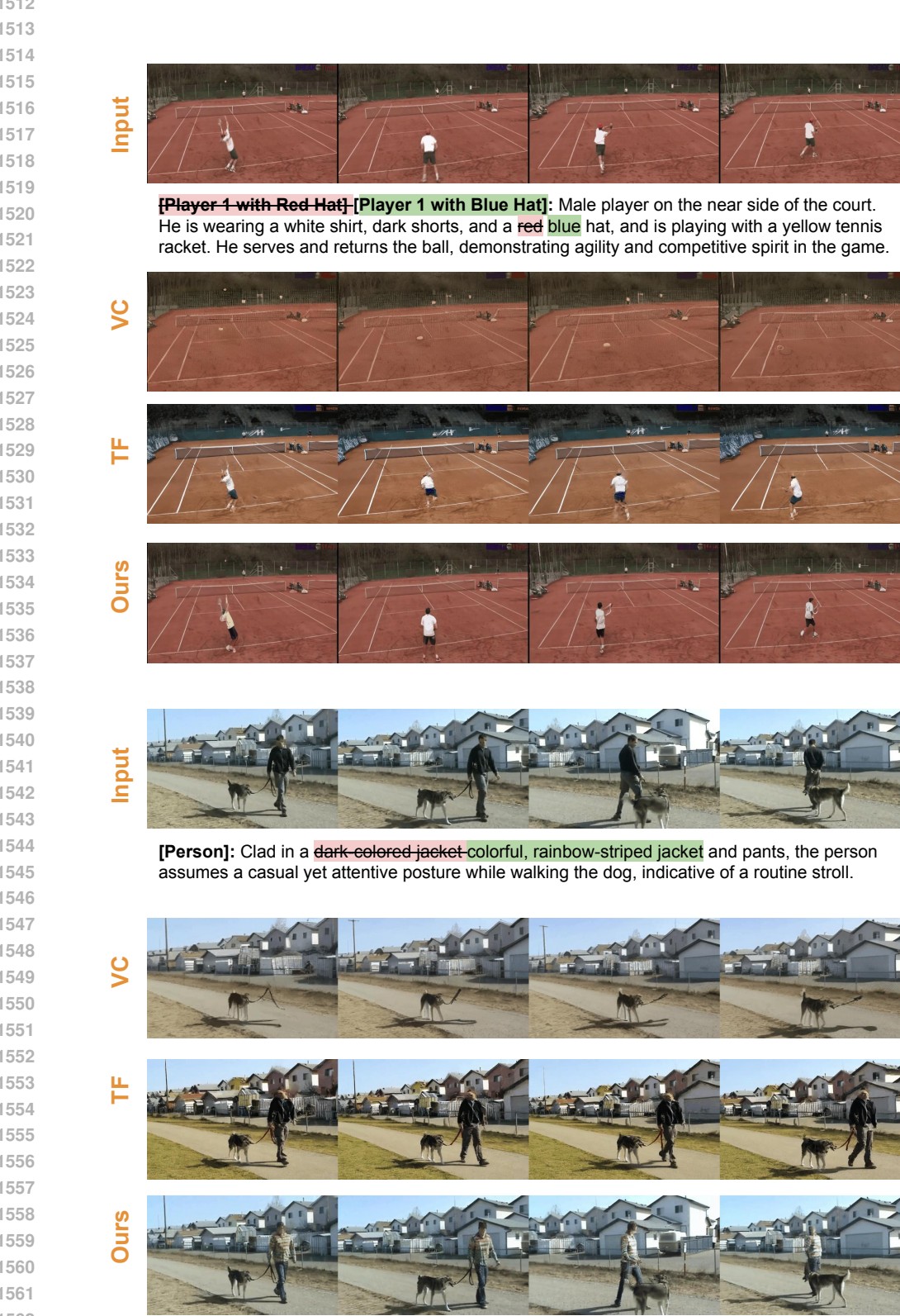

Figure 15: More visualization of **editing** video objects

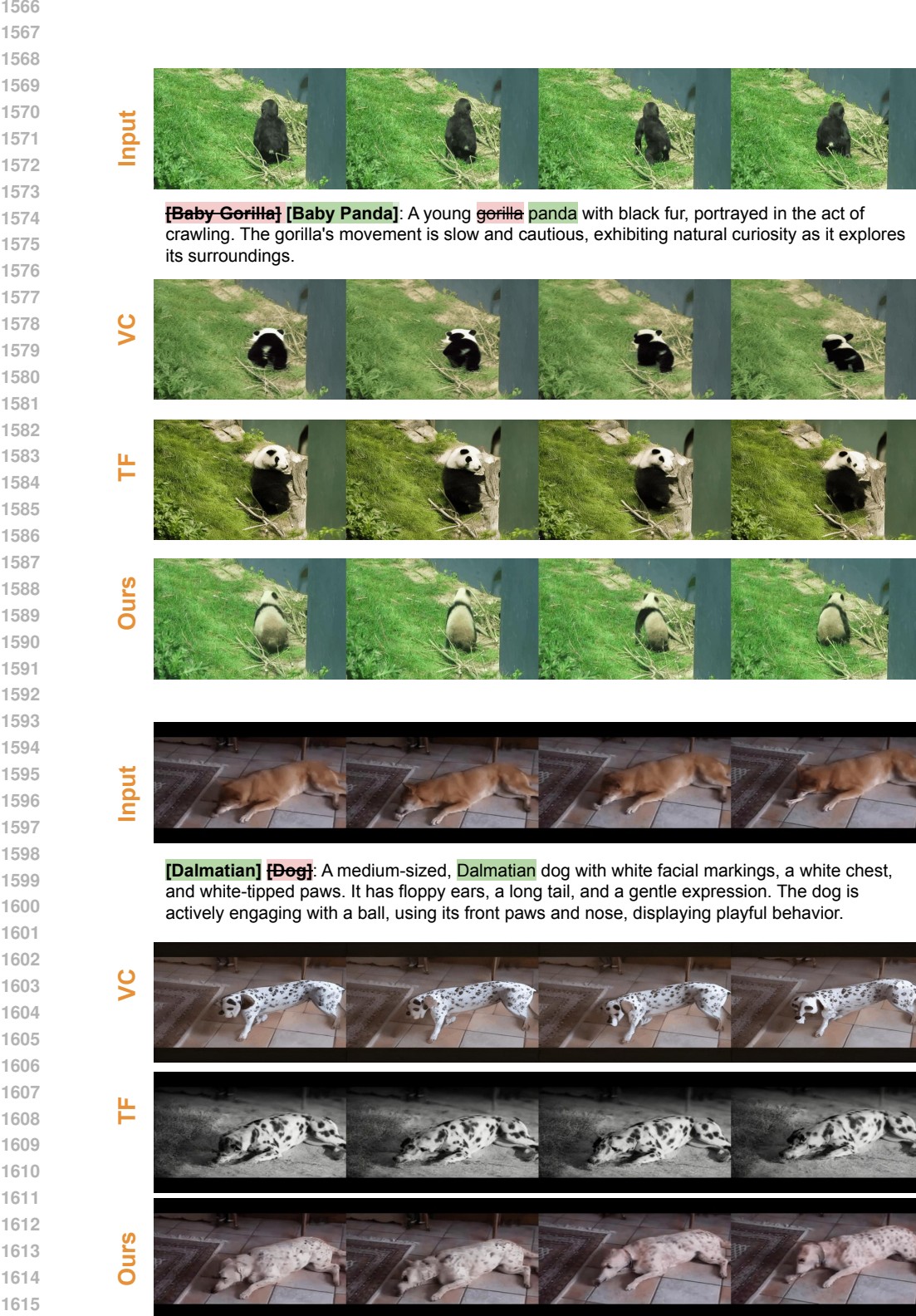

Figure 16: More visualization of **editing** video objects

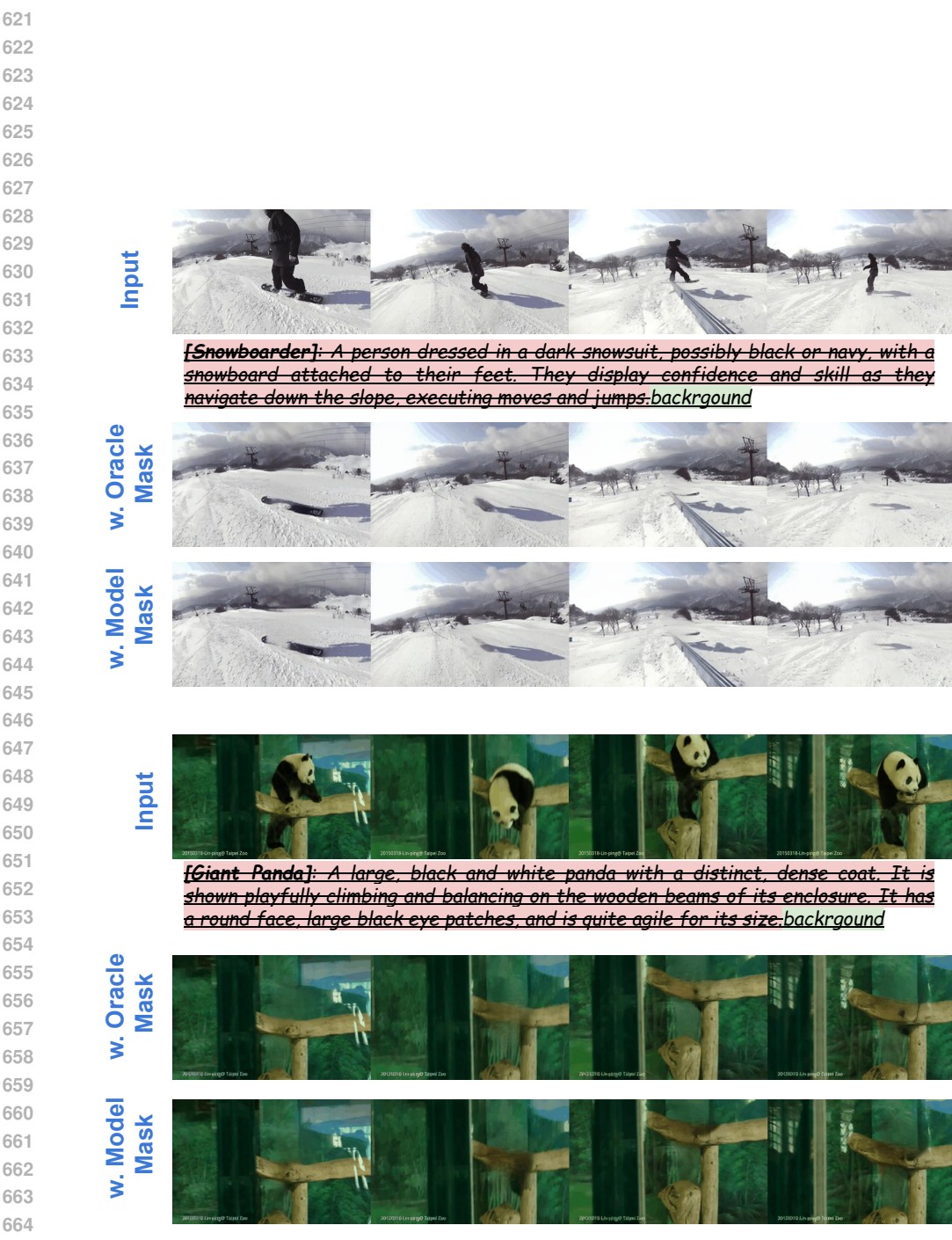

Figure 17: More visualization of **removing** video objects

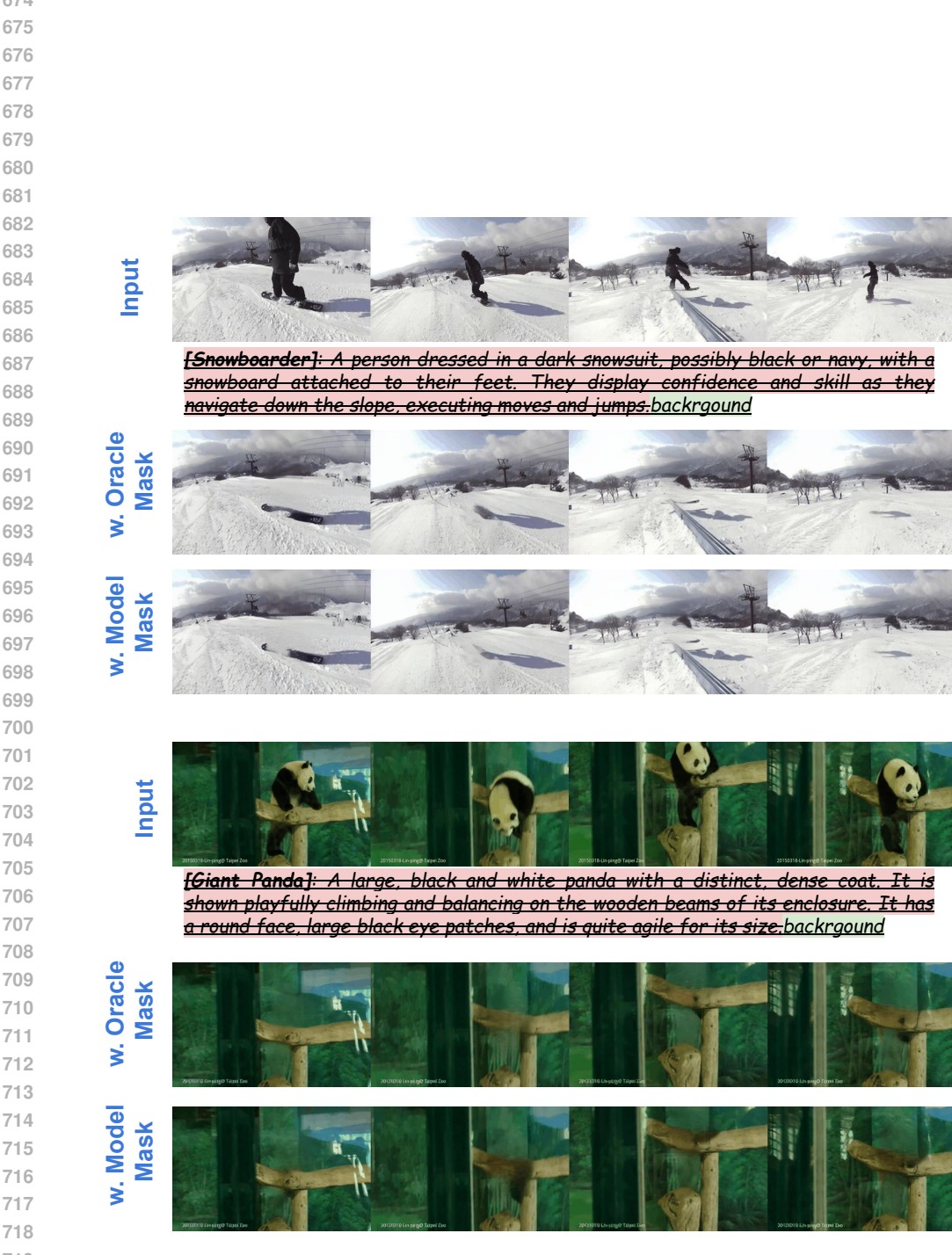

Figure 18: More visualization of **removing** video objects

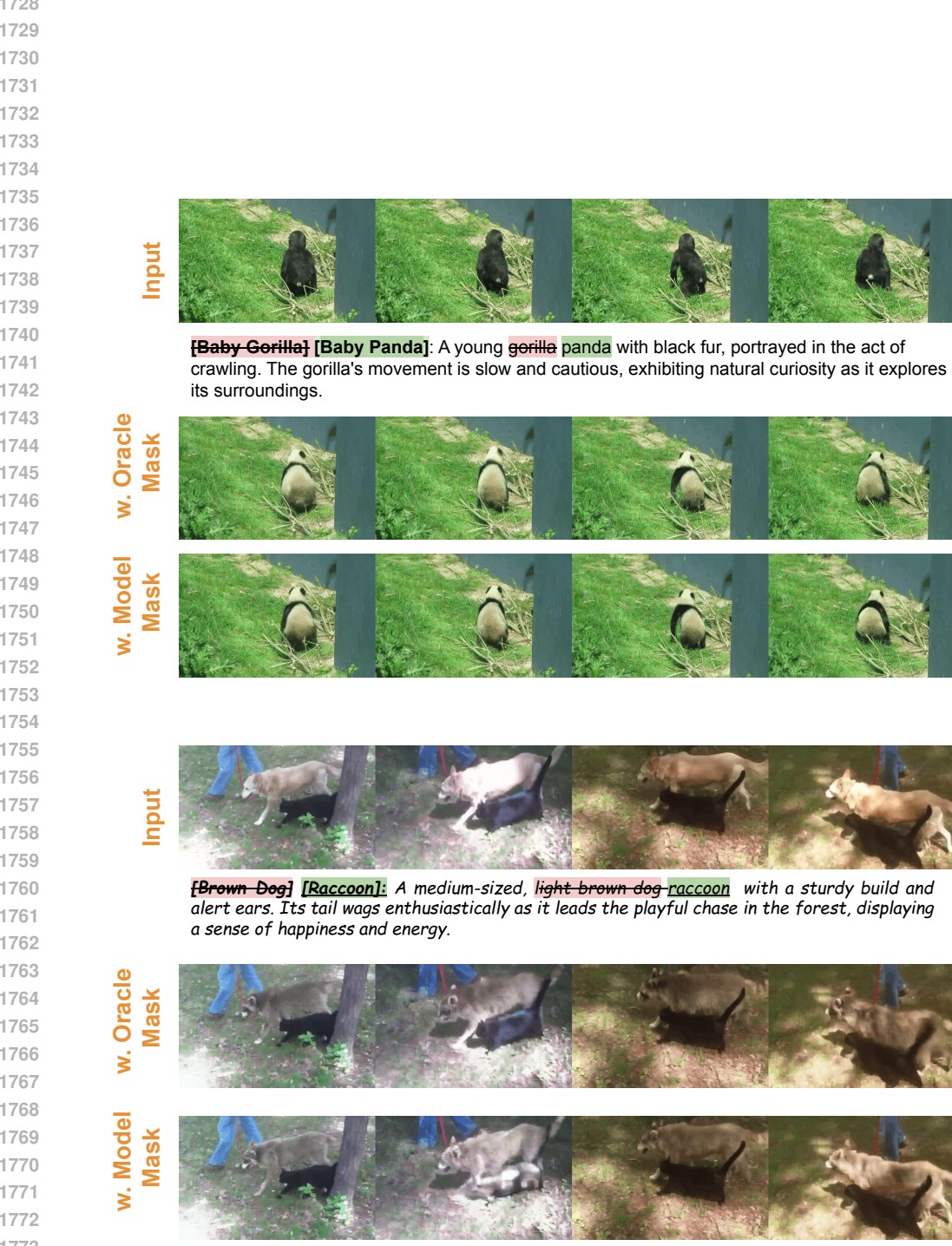

Figure 19: More visualization of **editing** video objects

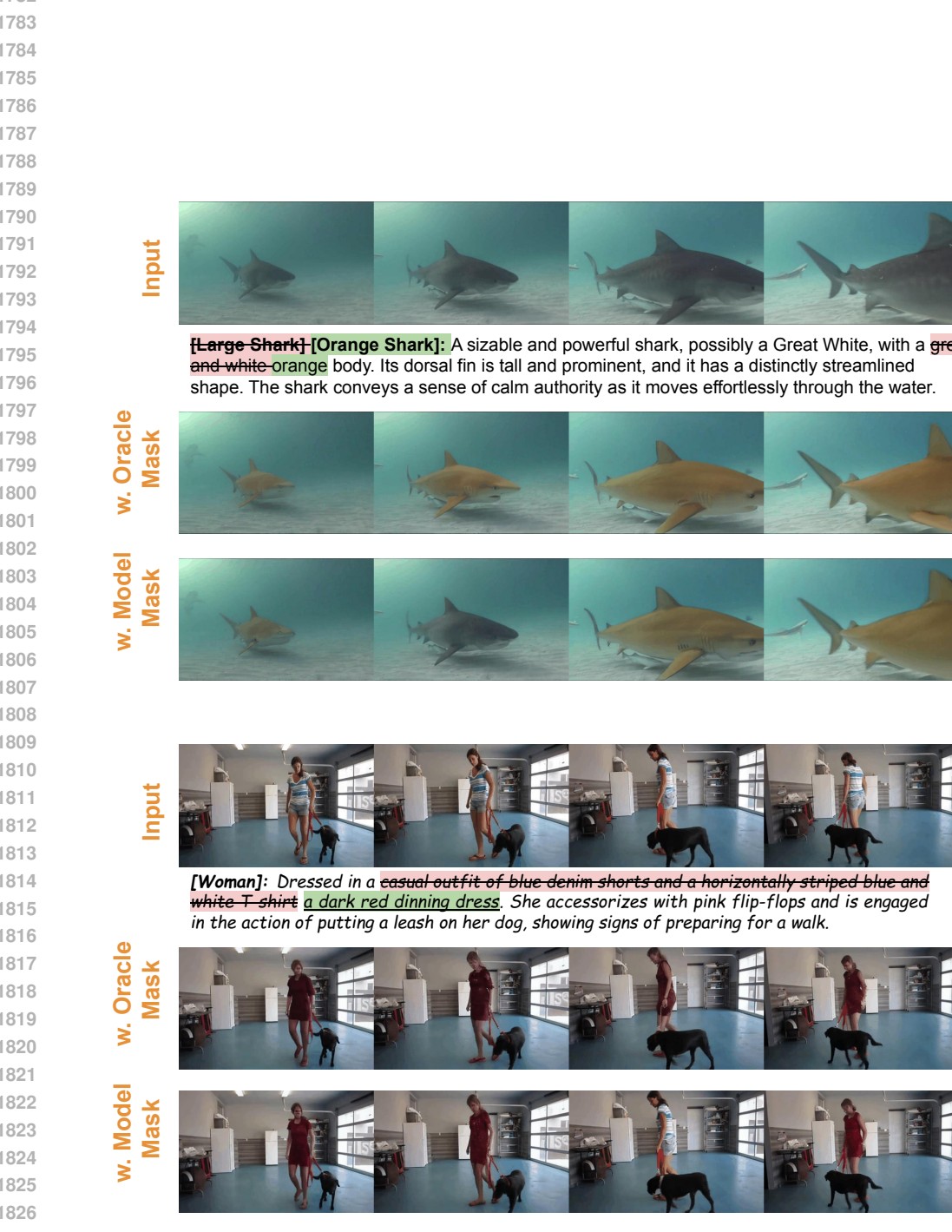

Figure 20: More visualization of **editing** video objects

**Input**

*+ [Surfer]: An athlete wearing a dark wetsuit, possibly black or navy, showcasing talent in balancing and steering on the waves. Their stance is wide and steady, knees bent, arms outstretched for balance, and their posture exuding confidence.*

**w. Oracle Planning**

**w. Model Planning**

**Input**

*+ [Giant Panda]: A large, black and white panda with a distinct, dense coat. It is shown playfully climbing and balancing on the wooden beams of its enclosure. It has a round face, large black eye patches, and is quite agile for its size.*

**w. Oracle Planning**

**w. Model Planning**

Figure 21: More visualization of **adding** video objects

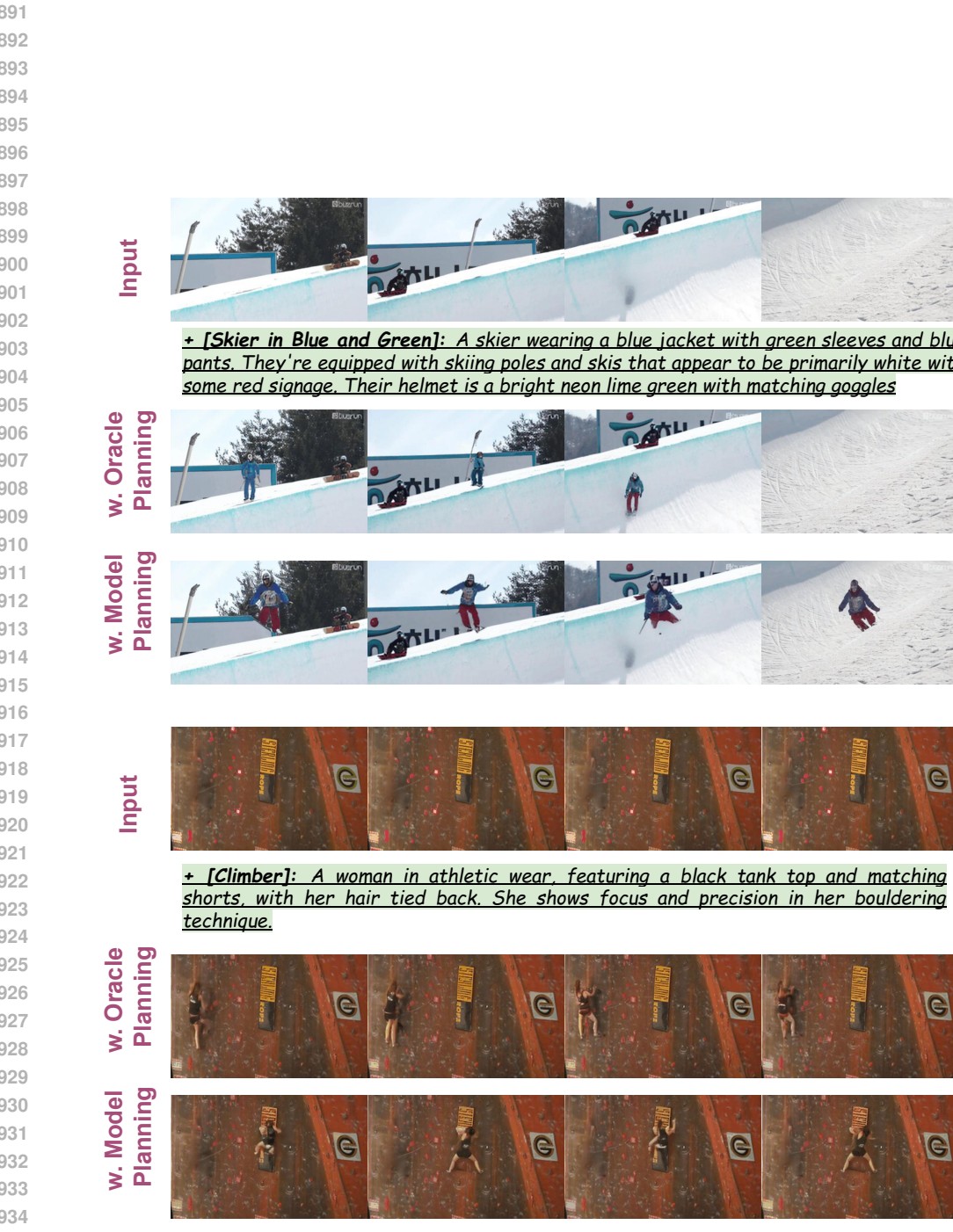

Figure 22: More visualization of **adding** video objects

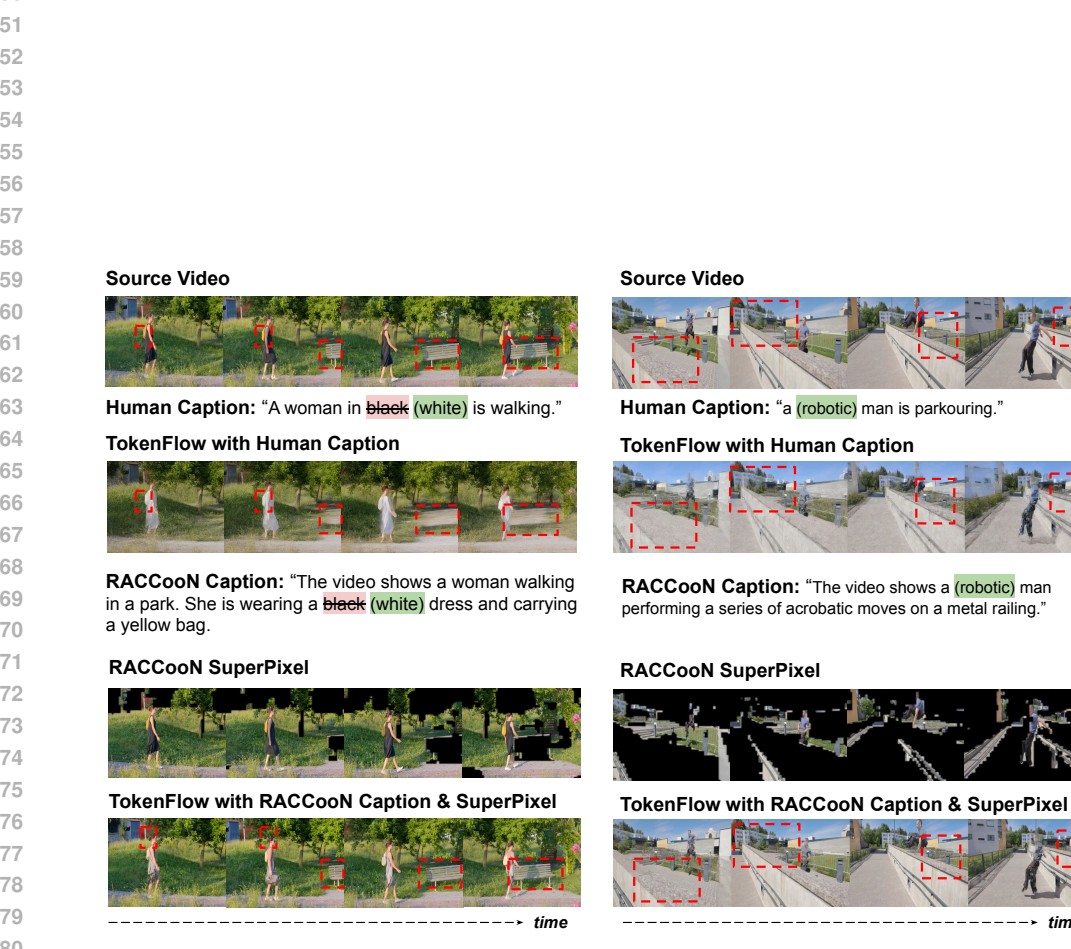

Figure 23: **Visualization of text-based video editing. The edited words are marked with Red, and the target words are marked with Green. The RACCooN caption is selected from predicted dense captions. We highlight the region of interest with red dashed-line boxes for comparison.**

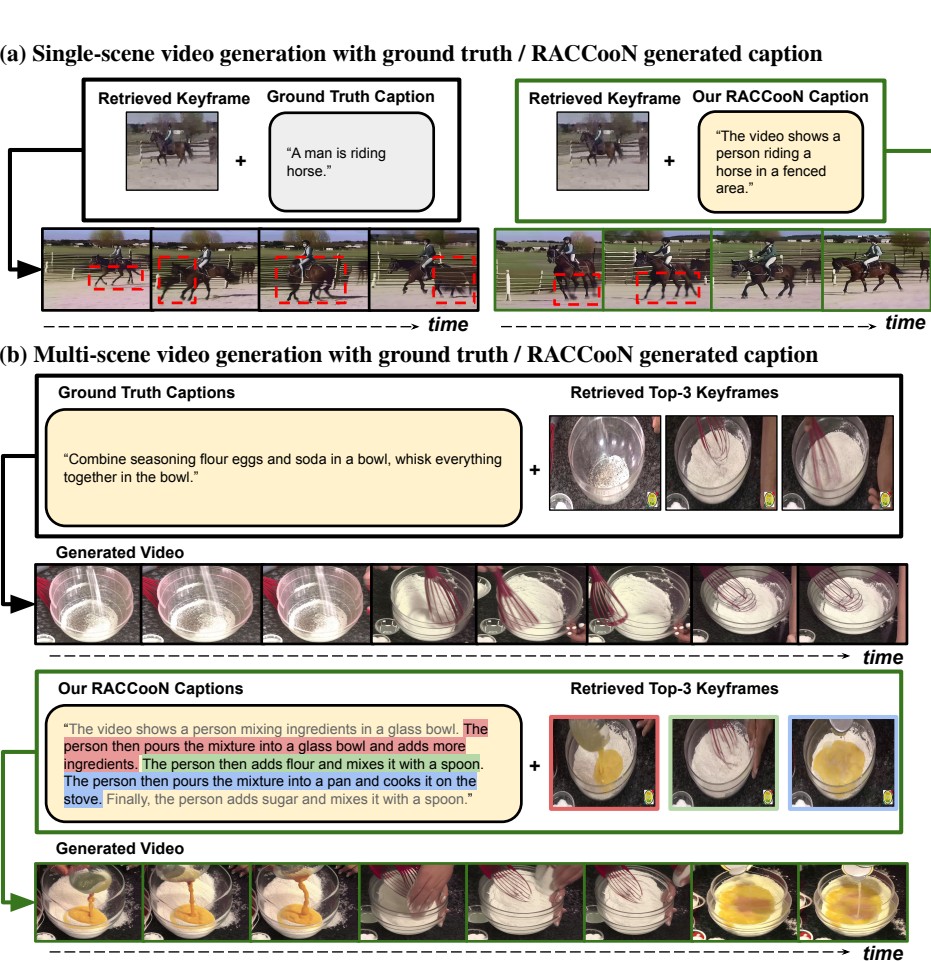

Figure 24: **Visualization of conditional video generation with VideoCrafter.** Top (a): we compare generation results conditional on different captions and with the **same** keyframe. Bottom (b): we leverage multiple keyframes retrieved by different captions to generate multi-scene video. We gray out captions that are not used for retrieval, and highlight captions used for keyframe retrieval. We highlight the region of distortion with red dashed-line boxes for detailed comparison.

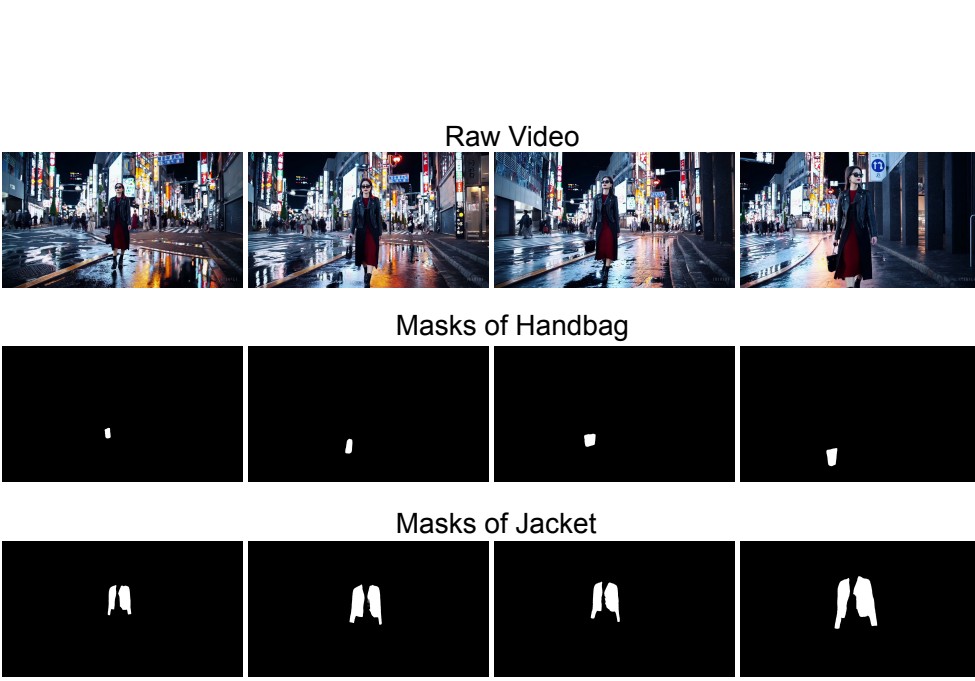

Figure 25: **Visualization of Masks.**

