# OpenReview forum: "RACCooN: A Versatile Instructional Video Editing Framework with Auto-Generated Narratives"
_ICLR.cc/2025/Conference — Submitted to ICLR 2025_

### Official Review · Reviewer_Bguv · 2024-10-30

**Soundness:** 3
**Presentation:** 2
**Contribution:** 3
**Rating:** 5
**Confidence:** 4

**Summary:**

This paper introduces RACCooN, a video-to-paragraph-to-video framework for enhancing video editing without the need for labor-intensive text prompts. It operates in two main stages: Video-to-Paragraph (V2P), which automatically generates natural language descriptions of video, and Paragraph-to-Video (P2V), where users can modify the descriptions to guide the video editing process.

**Strengths:**

1. The illustration figures are informative and help to understand the paper;
2. The experiment section provides detailed evaluations;
3. The supplementary materials help to understand the implementation and technical details.

**Weaknesses:**

1. The overall framework looks like a straightforward combination of a video understanding MLLM and a video editing diffusion model, and most of the techniques have been intensively explored by previous literature (e.g., the instruction tuning and training of the inpainting model). Thus, the core technical contribution is questionable.

2. I don't follow the formulation of the proposed instruction-based video editing task. Why it would be better to edit a video with intermediate text descriptions as a proxy, compared to directly providing pixel-level instruction?

3. For video editing evaluation, how are the videos selected, and how large is the evaluation set? Also, why it's necessary to evaluate PSNR and SSIM for video editing?

4. As an important component, there lacks an in-depth ablation study on the effects of the MGS pooling.

**Questions:**

1. What does "oracle mask" in ablation study mean?

2. How are the masks for video inpainting generated? Please consider attaching some results of the mask If it's produced by the LLM.

**Details Of Ethics Concerns:**

None.

---

> ### Author Response · Authors · 2024-11-21
> **Official Comment by Authors (1)**
>
> Dear Reviewer ```Bguv```,
>
>
> Thank you for your review and constructive comments. During the rebuttal period, we have made every effort to address your concerns. The detailed responses are below:
>
> ---
>
> > W1. The overall framework looks like a straightforward combination of a video understanding MLLM and a video editing diffusion model, and most of the techniques have been intensively explored by previous literature (e.g., the instruction tuning and training of the inpainting model). Thus, the core technical contribution is questionable.
>
> We respectfully disagree with the reviewer's argument that our novelty is marginal. The primary technical contributions of RACCooN do not lie in a naive concatenation of V2P and P2V methods. Instead, it is in the development of a versatile and user-friendly video editing framework. This framework supports **various video editing skills (remove/change/add) through a unified pipeline**. We highlight that the unique superiority of RACCooN **cannot be achieved by earlier SoTA models** (e.g., FateZero, TokenFlow, VideoComposer, LGVI) or by simply combining existing V2P and P2V methods (PGVL + SD-v2.0-inpainting).
>
> **Multi-granular Spatiotemporal Pooling (MGS Pooling):** Motivated by the limitations of existing Video-LLMs in capturing localized context (```Line 136-140```, ```144-146```), we propose a novel MGS Pooling method for MLLM (```Equation 1 and 2```). This design helps capture both holistic context and low-level details even in zero-shot settings (```Tables 7 and 8```) and shows superior performance over other fine-tuned baselines (```Tables 1 and 2```). Thus, the proposed MGS pooling also benefits automatic and objective-centric video data labeling, as recognized by the reviewer ```[HCrv]```.
>
> **Unified Video Editing Framework with Diverse Skills:** We have carefully designed to incorporate various editing skills (remove/change/add) into a unified framework, which was not achieved in previous works. Furthermore, our ablation study in ```Table 4``` emphasizes the benefits of automatically describing detailed video content and layout predictions for video editing in a unified framework.
>
> ---
>
> > W2. I don't follow the formulation of the proposed instruction-based video editing task. Why it would be better to edit a video with intermediate text descriptions as a proxy, compared to directly providing pixel-level instruction?
>
> We appreciate the reviewer’s concern. While pixel-level instructions (e.g., masks, edge maps) offer precise control, they require significant manual effort from users, particularly for tasks like providing **per-frame layouts to add objects to a video**, which is an **impractical burden for most users**.
>
> Also, we kindly note that **LoVECon** in ```Table 3``` is a ControlNet-based video editing model that directly provides **pixel-level conditions/instructions to the model**. Our RACCooN demonstrates generally superior performance across the three video editing skills (add, remove, and change), further underscoring the effectiveness of our framework design.
>
> Our RACCooN framework addresses this by **introducing a unified pipeline** that leverages MLLMs to automatically generate detailed object captions, masks (via grounding modules), and per-frame layouts. These auto-generated texts and visual conditions allow users to interactively select the desired conditions, significantly reducing manual effort while supporting diverse video editing capabilities.
> In this case, RACCooN effectively handles diverse editing tasks (add, remove, change), where other baselines often struggle and only handle a specific skill. This success is due to our decomposition of the video editing process into:
> - **V2P:** Using MLLMs to generate diverse conditions for the diffusion model.
> - **P2V:** The diffusion model applies these conditions to generate the edited video.
>
>
> This structured approach ensures versatility and ease of use, making the proposed method more practical and effective than direct pixel-level editing.
>
> ---
>
> > W3. For video editing evaluation, how are the videos selected, and how large is the evaluation set? Also, why it's necessary to evaluate PSNR and SSIM for video editing?
>
> We describe the dataset details in ```Section 4.1``` (```Line 366-367```), where each task/skill (adding/remove/change) is tested on 60 examples, so in total, we test all baseline methods and our RACCooN with 180 examples.
> In ```Table 3```, we test PSNR and SSIM for video object removal and addition tasks, following prior work [1], as we have “Ground Truth” videos for removing/adding objects in our dataset. Those inpainted videos are generated by a model and selected by humans to ensure GT quality. These metrics reflect the global similarity between generated videos and GT videos, it is crucial since we want to keep the background unchanged for inserting/deleting objects in a given video.
>
> [1] Towards Language-Driven Video Inpainting via Multimodal Large Language Models. CVPR24.

---

> ### Author Response · Authors · 2024-11-21
> **Official Comment by Authors (2)**
>
> > W4. As an important component, there lacks an in-depth ablation study on the effects of the MGS pooling.
>
> We politely correct the reviewer’s misunderstanding; we have **already provided the ablation of the proposed multi-granular pooling** strategy in our original submission. As shown in ```Table 8``` in the submission, these variants of RACCooN achieve improved performance against the best-performing baseline, PG-VL, but are often suboptimal since they focus on regional information and cannot contain temporal information of the videos. Unlike these image-based segmentation methods, SAM-Track generates coherent masks of observed objects over successive frames in videos, by adopting multiple additional pre-trained modules, including GroundingDino and AOT. We therefore adopt SAM-Track to initialize superpixels in videos and conduct overlapping k-means clustering (k = 25). Here, we observe that RACCooN with SAM-Track superpixel initialization achieves reasonable performance, and is beneficial for video editing tasks. It enables the model to coherently edit targeted regions in videos with edited keywords.
>
> ---
>
> > Q1. What does "oracle mask" in ablation study mean?
>
> Oracle masks indicate the ground-truth masks in the dataset.
>
> ---
>
> > Q2. How are the masks for video inpainting generated? Please consider attaching some results of the mask If it's produced by the LLM.
>
> In the object removal/change task, we use masks generated by the groundingDino+Sam-track module (```Lines 268-269```). We have also attached a visualization of masks in the revision (```Figure 25```).

---

> > ### Comment · Reviewer_Bguv · 2024-11-25
> > **Follow-up Questions**
> >
> > Dear authors,
> >
> > Thanks for the rebuttal. I still have some follow-up questions:
> >
> > 1. The language-based video editing is an interesting formulation. How does your framework tackle the hallucination issues that are common in MLLMs, e.g., the MLLMs produce descriptions that do not exist in the source video?
> >
> > 2. For the object-adding task, how are the masks generated?
> >
> > Thank you.

---

> > > ### Author Response · Authors · 2024-11-25
> > >
> > > >  The language-based video editing is an interesting formulation. How does your framework tackle the hallucination issues that are common in MLLMs, e.g., the MLLMs produce descriptions that do not exist in the source video?
> > >
> > > Thank you for the insightful question regarding hallucination issues in MLLMs. The authors agree that this challenge is indeed critical for ensuring the reliability of LLM-guided video editing and generation.
> > >
> > > From this perspective, we would like to emphasize that the proposed RACCooN framework has nice potential and functionality to mitigate hallucinations:
> > >
> > > - **Multi-Granular Pooling Strategy (MGS Pooling):** Recognizing the limitations of current Video-LLMs in capturing localized context, we propose a novel light-weighted MGS Pooling method specifically designed for MLLMs to effectively capture both holistic context and fine-grained details in the video. It enhances the MLLM in understanding video with less confusion/hallucinations compared to basic MLLMs with simple pooling that lacks structural video semantic and object-centric details. It is demonstrated by Table 3, that our framework outperforms naive baselines such as PG-Video-LLaVA + StableDiffusion 2.0-inpainting, reducing the risk of generating descriptions or edits inconsistent with the source video.
> > >
> > >
> > > - **User Interaction and Unified Editing Framework:** RACCooN framework supports direct user interaction, allowing users to refine and control the editing process effectively. This interactive functionality unifies multiple distinct video editing capabilities, ensuring the process remains interpretable and adaptable. Therefore, users can directly intervene to correct hallucinations or incorrectness (which is an unallowed function in most LLM-planning-based T2I/T2V generation/editing methods), preventing unintentional or hallucinated edits, further enhancing its practical usability and aligning with its 'user-friendly' design philosophy.
> > >
> > > We believe these two key components in RACCooN substantially alleviate hallucination risks while ensuring reliable and interpretable video editing capabilities.
> > >
> > > ---
> > >
> > > > For the object-adding task, how are the masks generated?
> > >
> > >
> > > For the object-adding task, masks are generated by the MLLM (V2P stage).
> > >
> > > The MLLM inputs the video and user instructions (e.g., "Add a dog to this video") to output bounding box coordinates in a (x_min, y_min, x_max, y_max) format for each frame (e.g., Frame1: [0.3, 0.4, 0.5, 0.6], Frame2: [0.5, 0.6, 0.7, 0.8]....). These coordinates are then converted into binary inpainting masks, targeting areas for the diffusion model to repaint.
> > >
> > > In this case, it automates mask generation for the object-adding, significantly reducing users’ labor to provide frame-wise mask conditions for adding an object not existing in the video.
> > >
> > > ---
> > >
> > >
> > > Thanks for your response, and we hope these additional clarifications could address your remaining questions.

---

> ### Author Response · Authors · 2024-12-01
> **We have less than two days left in the discussion period.**
>
> Dear Reviewer Bguv,
>
> We sincerely appreciate your efforts in reviewing our paper and your constructive comments. Since there are less than two days left in the discussion period, could you please read our responses to check if your concerns are clearly addressed? We believe that our responses resolved all your concerns.
>
> We understand that the criteria for rating a paper can sometimes be subjective; however, if you agree that our work does not have remaining major concerns, we would like to kindly suggest your re-evaluation of the initial rating of this submission.
>
> Please let us know if you have any remaining questions, and we will be more than happy to address them.
>
> Best,
> Authors

---

### Official Review · Reviewer_YJ8i · 2024-11-02

**Soundness:** 2
**Presentation:** 3
**Contribution:** 2
**Rating:** 5
**Confidence:** 5

**Summary:**

This work presents RACCooN, a video-to-paragraph-to-video generative framework designed to simplify video editing by automatically generating structured textual descriptions of video scenes. The process involves two main stages: V2P, which captures both the overall context and detailed object information, and P2V, where users can refine these descriptions to direct modifications to the video.

**Strengths:**

+ The paper is well-written and easy to understand, and the research questions are quite interesting.
+ The experimental results are relatively sufficient.

**Weaknesses:**

- The research goal of this work is not focused. The original objective is to address video editing, but in practice, it is solving the problem of video captioning.
- The method lacks some technical innovations. The approach to video editing can be summarized as text-based video inpainting, but it seems that there is no focused and innovative design specifically tailored to this goal.

**Questions:**

+ Calculating complexity. Video editing tasks can be very simple, but RACCooN complicates this process. Why is a caption model needed as a preprocessing step?
+ I think this work lacks two important baselines: 1) annotating the training set using the VLM from this paper; 2) annotating the training set using other state-of-the-art methods like GPT-4V. Then, we can train a model using Tex+Mask+Video. I believe the performance won't be worse, and this result could likely demonstrate that we do not need the V2P process.
+ Regarding multi-granular pooling, these three dimensions appear to be straightforward and commonly used and are insufficient to support the technical innovations presented in this work.

---

> ### Author Response · Authors · 2024-11-21
> **Official Comment by Authors (1)**
>
> Dear Reviewer ```YJ8i```,
>
>
> Thank you for your review and constructive comments. During the rebuttal period, we have made every effort to address your concerns. The detailed responses are below:
>
> ---
>
> > Q1. Video editing tasks can be very simple, but RACCooN complicates this process. Why is a caption model needed as a preprocessing step?
>
> In this rebuttal, we first respectfully correct the reviewer’s misunderstanding about the complexity of video editing tasks. Video editing is intuitively simple from a human perspective; however, it is well known that **skills to facilitate various types of editing capabilities,** such as object addition, object removal (inpainting), changing attributes, scene transition, and motion change, are **totally independent and hard to incorporate into a single framework** even **at image-level** [1].
>
> Therefore, the incorporation of the proposed video-to-paragraph (V2P) framework in RACCooN is crucial to address the inherent complexities of video editing, where tasks often require detailed prompts and robust scene understanding. RACCooN’s video-to-paragraph (V2P) framework is specifically designed to tackle these challenges effectively, leveraging an LLM as a ‘controller’ to generate conditions to cover as diverse skills as possible.
>
> - **Addressing Vague Prompts and Enhancing Scene Understanding:** Video editing typically relies on user-provided prompts, which are often ambiguous or high-level, making it difficult for models to infer precise editing actions. Existing methods like VideoComposer, LGVI, and TokenFlow struggle with dynamic scenes and visually complex videos.
>
>   - **Object Editing in Dynamic Scenes/Movements:** As shown in ```Figure 5 (bottom right)```, TokenFlow and VideoComposer fail to consistently edit the attribute of a *"Basketball Player in White"*, leading to inaccurate editing results when changing their clothing to "a blue shirt and white shorts." RACCooN, in contrast, excels by leveraging enhanced scene understanding.
>   - **Object Removal in Confusing Scene:** Similarly, in ```Figure 5 (top left)```, RACCooN successfully removes the *"Climber in Green,"* avoiding residual artifacts like hands or legs, where other models struggle due to visual confusion.
>
>   These examples highlight RACCooN’s ability to handle fine-grained editing requests, thanks to its improved understanding of complex actions and dynamic objects.
>
> - **Facilitating Various Video Editing Skills within a Unified Framework via Human Interactions:** The auto-generated video paragraphs act as an interpretable bridge, enabling users to specify and refine edits at a granular level. This ensures precise alignment of user instructions with detailed video descriptions, making various ranges of editing skills (object addition, object removal (inpainting), and changing attributes) more intuitive.
>
> - **Improving Training and Output Quality:** Caption models enrich training signals with context-rich descriptions, mitigating issues caused by vague training data. This leads to high-quality, user-aligned video edits that deliver superior fidelity and relevance.
>
> The V2P stage in RACCooN is not a complication but **a vital component addressing challenges** such as vague prompts, complex content, and the need for precise, user-aligned edits. Enhancing the understanding, interpretation, and execution of user instructions ensures high-fidelity outputs. Moreover, RACCooN's lightweight superpixel predictor enables efficient visual grounding and paragraph generation within marginal seconds, maintaining a seamless user experience during video editing.
>
> [1] Wei et al., OmniEdit: Building Image Editing Generalist Models Through Specialist Supervision, aiXiv 2411.07199
>
>
> ---
>
> > W1. The research goal of this work is not focused. The original objective is to address video editing, but in practice, it is solving the problem of video captioning.
>
> We would like to kindly address the reviewer’s misunderstanding regarding the research focus of our paper. We emphasize that enhancing the performance of **video generation or editing models through planning** is a well-established and impactful area of research. In line with this, our primary objective is to enable diverse video editing capabilities, including Remove, Add, and Change operations, by leveraging auto-generated narratives. Importantly, our approach achieves this **without requiring extensive user input for planning**. This makes our method both practical and accessible, aligning with the broader goal of advancing video editing methodologies.

---

> ### Author Response · Authors · 2024-11-21
> **Official Comment by Authors (2)**
>
> > W2. The method lacks some technical innovations. The approach to video editing can be summarized as text-based video inpainting, but it seems that there is no focused and innovative design specifically tailored to this goal.
>
> We respectfully disagree with the reviewer's argument that our novelty is marginal. We would like to emphasize two key points in response to your concern about the lack of technological innovation in our approach:
>
> - **Broader Research Contributions:** We believe it is essential to highlight that scientific contributions in the research field should not be assessed solely on the basis of technical novelty. Technical contribution is **one dimension of a research contribution**. Equally important are framework contributions, which involve developing effective methodologies to address key challenges, and dataset contributions, which provide the community with valuable resources grounded in solid motivations and observations. Our work contributes meaningfully to these areas, offering a comprehensive framework for text-based video editing and introducing a dataset tailored to evaluate and advance such approaches. Specifically, RACCooN additionally offers solid and meaningful contributions:
>   - **Framework Contribution:** A user-friendly, unified framework that automatically generates detailed, object-centric video descriptions and layout plans tailored to editing objectives, offering capabilities beyond simple model combinations.
>   - **Training and Dataset Contribution:** The VPLM dataset (7.2K video paragraphs, 5.5K object-level caption-mask pairs) supports high-quality Video-to-Paragraph (V2P) and Paragraph-to-Video (P2V) transformations, setting new standards for video editing tasks.
> - **Concrete Technical Innovations:** At the same time, we would politely remind you that the technical contributions of RACCooN are rooted in well-defined motivations and connections that are specifically designed to address the challenges of video editing. These innovations extend beyond mere text-based video inpainting, enabling a more comprehensive and tailored approach.
>
> ---
>
> > Q2. I think this work lacks two important baselines: 1) annotating the training set using the VLM from this paper; 2) annotating the training set using other state-of-the-art methods like GPT-4V. Then, we can train a model using Tex+Mask+Video. I believe the performance won't be worse, and this result could likely demonstrate that we do not need the V2P process.
>
> In our understanding, the suggested training design implies fine-tuning the video editing model with VLM-generated prompts and masks. If it is correct,
> - The suggested design is technically the same as conventional video editing models. That is, these models lack the capability to **streamline the video editing process** and **minimize user effort**, which is the primary objective and practical advantage of RACCooN. By "user-friendly," we specifically mean that RACCooN automates much of the detail recognition and editing process. Users can edit videos simply based on auto-generated text descriptions, **eliminating the need for extensive manual intervention or technical expertise**. This unique capability sets our framework apart and aligns with our goal of simplifying video editing for a broader range of users.
> - Moreover, the suggested approach of fine-tuning video diffusion models using VLM-generated data is **not as straightforward as it might seem**. It involves significant challenges, including complex and scalable data collection (e.g., how do we ‘annotate’ video data using VLM), filtering, and the careful design of training processes to ensure quality and consistency. These requirements **further emphasize the importance of our dataset and training contributions of RACCooN**, which are specifically designed to address these challenges and provide a robust foundation for effective video editing.
>
> We appreciate the reviewers' questions and are happy to provide further clarification or address any potential misinterpretation of the original question in our rebuttal.
>
> ---
>
> > Q3. Regarding multi-granular pooling, these three dimensions appear to be straightforward and commonly used and are insufficient to support the technical innovations presented in this work.
>
> We respectfully disagree with the argument that our multi-granular pooling is straightforward and commonly used. Our approach introduces a novel aspect: the use of **superpixel-based overlapped k-means clustering**, which effectively groups relevant spatiotemporal pixels in a way that **has not been explored in previous research**. This innovation serves as the foundation for our multi-granular pooling strategy, which is distinctly novel and provides a unique contribution to the field, evident in the ablation in ```Table 8```. By leveraging this method, we enhance the ability to capture and represent spatiotemporal features, addressing a gap that previous work has not tackled.

---

> ### Author Response · Authors · 2024-11-25
> **A gentle reminder**
>
> Dear Reviewer YJ8i,
>
> Thank you for your effort in reviewing our paper. We kindly notify you that the end of the discussion stage is approaching. Could you please read our responses to check if your concerns are clearly addressed? During the rebuttal period, we made every effort to address your concerns faithfully:
>
> - We have corrected the reviewer's misunderstanding about why video RACCooN benefits from complicating this process to address video editing tasks. We respectfully emphasize that it is well known that **skills to facilitate various types of video editing capabilities are totally independent and hard to incorporate into a single framework**. To the best of our knowledge, no existing framework achieves such a generalizable and robust video editing framework.
> - We have corrected the reviewer's misunderstanding about the research goal of RACCooN. RACCooN's research goal is **solid and consistent**: Build a unified framework incorporating diverse video editing capabilities, including Remove, Add, and Change operations, by **leveraging auto-generated narratives and layout planning**, without requiring extensive user input for planning.
> - We have addressed the reviewer's concern regarding technical contribution. We politely emphasize our broader research contributions and solid technical innovations demonstrated by extensive quantitative and qualitative evaluations and ablations.
> - We have clarified a slight misunderstanding of baselines and the objective of RACCooN. We would be happy to provide further clarification or discuss this with the reviewer.
>
>
>
> ---
>
> Thank you for your time and effort in reviewing our paper and for your constructive feedback, which has significantly contributed to improving our work. We hope the added clarifications and the revised submission address your concerns and kindly request to further **reconsider the rating/scoring**. We are happy to provide further details or results if needed.
>
> Warm Regards,
> Authors

---

> ### Author Response · Authors · 2024-12-01
> **We have less than two days left in the discussion period.**
>
> Dear Reviewer YJ8i,
>
> We sincerely appreciate your efforts in reviewing our paper and your constructive comments. Since there are less than two days left in the discussion period, could you please read our responses to check if your concerns are clearly addressed? We believe that our responses resolved all your concerns.
>
> We understand that the criteria for rating a paper can sometimes be subjective; however, if you agree that our work does not have remaining major concerns, we would like to kindly suggest your re-evaluation of the initial rating of this submission.
>
> Please let us know if you have any remaining questions, and we will be more than happy to address them.
>
> Best,
> Authors

---

### Official Review · Reviewer_h1ZE · 2024-11-03

**Soundness:** 3
**Presentation:** 3
**Contribution:** 2
**Rating:** 5
**Confidence:** 4

**Summary:**

This work introduces a novel approach for video editing that includes both Video-to-Paragraph (V2P) and Paragraph-to-Video (P2V) methodologies. Initially, the V2P method converts video content into a descriptive paragraph. Subsequently, users can edit this textual representation to facilitate video object addition, removal, and modification. Experimental results demonstrate that our approach outperforms existing state-of-the-art methods in the field.

**Strengths:**

- The proposed Video-to-Paragraph (V2P) method serves as an effective video captioner, capable of capturing multi-granular spatiotemporal features.
- This framework achieves state-of-the-art performance in video editing tasks.
- A novel dataset is introduced, which can be used as a benchmark for video editing, which contains 7.2K high-quality detailed video paragraphs and 5.5K object-level detailed caption-mask pairs.

**Weaknesses:**

1. The proposed editing approach heavily relies on the accuracy of the Video-to-Paragraph (V2P) method, leading to potentially unnatural modifications.
   - (1) For video object removal and modification, if the paragraph does not mention a specific object (e.g., in the description of Figure 4, the absence of the female character’s earrings), it raises the question of how to effectively remove or modify that object.
   - (2) While adding objects does not depend on the paragraph's accuracy, it is unclear how to control the timing of additions through text. The mechanism for locating the spatiotemporal position of the mask for object addition is lacking in detail. Is the layout determined using a large language model (LLM), or is there another implementation?
   - (3) The textual descriptions lack a temporal dimension, making it difficult to achieve fine-grained control over video content, such as adding objects at specific time intervals or altering object attributes over time.
   - (4) There is no comparison of the Video-to-Paragraph method with more advanced video LLM techniques. In the single object prediction task, there is a lack of comparison with state-of-the-art methods like Video-Chat2 and Video-LLaVA.
   - (5) While the editing pipeline boasts high accuracy, it is not user-friendly. Unlike methods such as Instruct Pix2Pix, which allow for straightforward edits, this approach requires modifications to the existing paragraph.


2. The multi-granular pooling strategy lacks ablation studies, which are necessary to evaluate its effectiveness.

Additonal discussion:
For image editing, the workflow of converting images to text and then modifying through text works well; however, video editing inherently requires temporal control, and the current approach lacks mechanisms for time-based edits and fine-grained temporal adjustments—an essential aspect of video manipulation.

The pipeline could benefit from further streamlining. Ideally, users would simply input a high-level instruction, and the remainder of the editing process would be handled by the LLM, removing the need for users to read and modify the paragraph directly.

**Questions:**

See Weakness.

---

> ### Author Response · Authors · 2024-11-21
> **Official Comment by Authors (1)**
>
> Dear Reviewer ```h1ZE```,
>
>
> Thank you for your review and constructive comments. During the rebuttal period, we have made every effort to address your concerns. The detailed responses are below:
>
> ---
>
> > W2. The multi-granular pooling strategy lacks ablation studies.
>
> We politely correct the reviewer’s misunderstanding; we have **already provided the ablation of the proposed multi-granular pooling** strategy in our original submission. As shown in ```Table 8``` in the submission, these variants of RACCooN achieve improved performance against the best-performing baseline, PG-VL, but are often suboptimal since they focus on regional information and cannot contain temporal information of the videos. Unlike these image-based segmentation methods, SAM-Track generates coherent masks of observed objects over successive frames in videos, by adopting multiple additional pre-trained modules, including GroundingDino and AOT. We therefore adopt SAM-Track to initialize superpixels in videos and conduct overlapping k-means clustering (k = 25). Here, we observe that RACCooN with SAM-Track superpixel initialization achieves reasonable performance, and is beneficial for video editing tasks. It enables the model to coherently edit targeted regions in videos with edited keywords.
>
>
> ---
>
> > W1. RACCooN heavily relies on the accuracy of the Video-to-Paragraph (V2P) method, leading to potentially unnatural modifications.
>
> We politely disagree with the reviewer's two arguments that our RACCooN relies on the V2P method and leads to potentially unnatural modifications. We provide our detailed rebuttal regarding the reviewer’s questions or claims below:
>
> ---
>
> > W1-(1) If the paragraph does not mention a specific object, how can that object be effectively removed or modified?
>
> We appreciate the reviewers' meaningful questions regarding cases where specific objects are not described in the auto-generated prompts. We would politely emphasize that the framework **supports the effective removal** or modification of specific objects **even when the paragraph generated by V2P does not explicitly mention them**. The V2P process provides a list of candidate objects in the video that are fine-grained yet sufficiently significant or meaningful by analyzing the video content. This approach reduces the users’ efforts or burdens to make careful manual observations while still allowing flexibility in editing.
>
> For example,
> - for the **removal skills** of RACCooN, if the V2P stage does not capture fine-grained details like "earrings” that the user wants to remove, the user can directly provide prompts like “earrings” to a segmentation model to obtain the mask for P2V diffusion model for editing.
> - And for the **changing task** of RACCooN, users can simply prompt the V2P model with a prompt like “Describe the earrings”. Then the MLLM can generate detailed object descriptions for video editing. Those detailed descriptions play a critical role in preserving essential objects or attributes during the P2V process for editing.
>
> ---
>
> > W1-(2) It is unclear how to control the timing of additions through text. The mechanism for locating the spatiotemporal position of the mask for object addition is lacking in detail.
>
> We appreciate the reviewer’s suggestion to provide a more detailed explanation of the process for adding objects and predicting layouts in RACCooN.
>
> First, we would like to address a slight misunderstanding. The capability of the video-to-paragraph (V2P) stage directly **influences the performance of video editing tasks**, including ‘adding objects’. As clarified in our response to the review ```W1-(1)```, the detailed descriptions of existing content, automatically generated by the V2P stage, play a critical role in preserving essential objects and attributes during the inpainting process. This impact is evident in ```Table 4``` of our submission, where we validate the effectiveness of the V2P framework. By replacing auto-generated detailed descriptions with short captions, we observe a significant enhancement in generation quality, resulting in a **+14.4%** relative improvement in FVD, as precise and accurate details are crucial for maintaining quality.
>
>
> Regarding the mechanism for determining the placement of objects to be added, we clarify in ```Lines 179–180```: *“Note that for adding object tasks, if users do not provide layout information for the objects they want to add, RACCooN can predict the target layout in each frame.”* Additionally, in ```Lines 265–267```: *“the MLLM in the V2P process provides not only detailed descriptions but also frame-wise placement suggestions for new objects.”*
>
> (continued)

---

> ### Author Response · Authors · 2024-11-21
> **Official Comment by Authors (2)**
>
> This process is also illustrated in ```Figure 2```, which outlines the workflow as follows:
> - **1. User Edit**: The user provides instructions on how to add a specific object.
> - **2. MLLM Output**: The finetuned MLLM in the V2P process generates fine-grained video descriptions along with frame-wise bounding box suggestions for new objects. For example, “Layouts of <Obj> to be added: {Frame 1: ```[0.2, 0.0, 0.5, 0.7]```, Frame 2: ```[0.2, 0.1, 0.4, 0.65]```, …” specifies the layout for each frame, where ```[x1, y1, x2, y2]``` represents the top-left and bottom-right corners of the bounding box, with coordinates normalized to the range [0, 1] (as shown in the yellow box in Figure 2, top right).
> - **3. Video editing**: Generate videos based on the MLLM-generated output, including the frame-wise layout of the object to be added.
>
> We acknowledge and appreciate the reviewer’s valuable suggestion to provide a more detailed explanation regarding the process of adding objects in video editing. Indeed, adding objects is a unique task, distinct from removing objects or changing attributes. Unlike the latter scenarios, where the target objects are already present in the initial video, adding objects involves introducing entirely new elements, which necessitates a slightly different editing process.
>
> In response to this feedback, we have **integrated the requested details about the process of adding objects in our revision**. This includes clarifications on how layouts are predicted for new objects, how frame-wise bounding boxes are generated, and how these elements interact with the inpainting and refinement steps. We hope these additions will address the reviewer’s concerns and provide a clearer understanding of this unique and impactful functionality. (Please see ```Lines 260-277``` in our revision).
>
> Thank you again for your constructive feedback.
>
> ---
>
>
> > W1-(3) & Additional discussion 1: The textual descriptions lack a temporal dimension, making it difficult to achieve fine-grained control over video content, such as adding objects at specific time intervals or altering object attributes over time. & additional discussion regarding temporal adjustments in video editing.
>
> We acknowledge the concern regarding the absence of a temporal dimension in the textual descriptions generated by our framework, which could present challenges in achieving fine-grained control over video content, such as adding objects at specific time intervals or dynamically altering object attributes over time.
>
> The field of video editing includes **various types of editing**; *temporal-dependent alteration of object attributes* is another important line of video editing, different from our major video editing tasks, such as *high-level edits to video content* - modifying object properties and adding or removing specific objects: While temporal control is undoubtedly critical for certain video manipulation applications, **our work represents a distinct and important line of research that differs from motion-focused video editing**. In addition, we would like to emphasize two critical points:
>
> - **No existing baselines are capable of such general-purpose video editing & conditional generation capability:** Our setting is sufficiently challenging as, to the best of our knowledge, most strong video editing methods would **fail to handle these diverse editing skills in a single framework** (e.g., InstructVid2Vid, TokenFlow). Specifically, our framework prioritizes enabling users to make high-level edits to video content, such as modifying object properties and adding or removing specific objects, which aligns more closely with general editing use cases. Exploring advanced temporal editing capabilities could be a valuable extension of this work in the future, but it is clearly outside the current scope of our system.
>
> - **The promising potential of Raccoon’s fine-grained planning to address temporal adjustment:** We believe that extending our framework to incorporate temporal planning and control represents an exciting and meaningful direction for future research. By advancing RACCooN's ability to handle time-dependent edits, the framework could achieve even greater levels of controllability and versatility, unlocking new possibilities in video editing and expanding its applicability to a broader range of scenarios. This advancement would not only enhance the precision of edits but also solidify RACCooN's position as a comprehensive tool for complex video editing tasks.

---

> ### Author Response · Authors · 2024-11-21
> **Official Comment by Authors (3)**
>
> > W1-(4) No comparison of the Video-to-Paragraph method with more advanced video LLM techniques, Video-Chat2 and Video-LLaVA.
>
> We politely clarify that the comparison of the Video-to-Paragraph method with more advanced video LLM techniques, such as Video-Chat2 and Video-LLaVA, is **clearly beyond the scope of our research**. Our primary focus is on **video editing capabilities** rather than the specific architecture or advancements of the underlying LLM backbone. The methodology we propose is designed to be **agnostic to the choice of LLM backbone**, allowing for flexibility and adaptability across different frameworks. As a result, we emphasize the applicability of our approach to video editing tasks rather than direct comparisons with advanced video LLM models.
>
> ---
>
> > W1-(5) The claim regarding the ‘user-friendliness’ of RACCooN, compared to other baselines allowing for direct visual editing, like InstructPix2Pix.
>
> We would respectfully correct a misunderstanding regarding the use of the term *‘user-friendliness’* in our submission. Unlike the case of InstructPix2Pix implementing ‘user-friendliness’ for direct visual editing via only user input textual instruction, the claim regarding the 'user-friendliness' of RACCooN refers to **its ability to simplify the video editing process with minimal user effort**.
>
> Particularly when compared to other baselines like InstructPix2Pix. By 'user-friendly,' we specifically mean that RACCooN automates much of the detail recognition in videos, allowing users to easily **select and edit** object candidates in videos through auto-generated text descriptions. This approach further reduces the user input, making it accessible to a broader range of users, including those without prior experience in video editing or advanced software tools. This emphasis on automation and simplicity enhances RACCooN's appeal as a convenient and efficient tool for video editing.
>
> In this rebuttal, we also show that our proposed RACCooN framework **outperforms strong instructional image/video editing models w.o. MLLM (MagicBrush / InstructVid2Vid) on several editing skills**. It further highlights our framework also demonstrated better effectiveness in handling diverse editing skills.
>
>
> Models | FVD | SSIM | PSNR
> |-|-|-|-|
> ```Remove Object```
> MagicBrush  |  1355.95  | 30.22 | 16.18
> instructVid2Vid | 1509.40 | 21.93 | 14.70
> **RACCooN** |  **162.03** | **84.38** | **30.34**
> ```Add Object```
> MagicBrush  | 1118.42 | 35.32 | 18.41
> InstructVi2Vid | 1466.44 | 21.91 | 14.68
> **RACCooN** | **415.82** | **77.81** | **23.38**
>
> [1] Magicbrush: A manually annotated dataset for instruction-guided image editing. CVPR24.
> [2] Consistent Video-to-Video Transfer Using Synthetic Dataset. ICLR24
>
> ---
>
> > Additional discussion 2: The pipeline could benefit from further streamlining. Ideally, users would simply input a high-level instruction, and the remainder of the editing process would be handled by the LLM, removing the need for users to read and modify the paragraph directly.
>
> We greatly appreciate the reviewer’s insightful suggestion regarding streamlining the pipeline by enabling users to input high-level instructions while delegating the remainder of the editing process entirely to the LLM, thereby eliminating the need for users to directly read or modify the generated paragraph. We agree that this approach represents an excellent direction for future improvement.
>
> Currently, our pipeline incorporates some level of user interaction to refine textual descriptions, ensuring precise control over the editing process. However, we recognize that minimizing user intervention by automating these steps would significantly enhance the overall user experience, making the system more intuitive and accessible.
>
> Incorporating such functionality aligns with our long-term vision for the RACCooN framework. By further integrating advanced LLM capabilities with video editing workflows, we aim to develop a fully automated and seamless editing process. This suggestion provides invaluable guidance for future iterations of our work and will undoubtedly help us refine and expand the capabilities of RACCooN to better serve its intended users.

---

> ### Author Response · Authors · 2024-11-25
> **A gentle reminder**
>
> Dear Reviewer h1ZE,
>
> Thank you for your effort in reviewing our paper. We kindly notify you that the end of the discussion stage is approaching. Could you please read our responses to check if your concerns are clearly addressed? During the rebuttal period, we made every effort to address your concerns faithfully:
>
> - We have addressed the concern regarding ablation studies of the multi-granular pooling strategy. We have already provided the ablation of the proposed multi-granular pooling strategy in our original submission (Table 8).
> - We have addressed the concern regarding the video editing case when the paragraph does not mention a specific object. The RACCooN framework supports the effective removal or modification of specific objects even when the paragraph generated by V2P does not explicitly mention them.
> - We have provided a detailed and comprehensive process on how to control the timing of additions through text.
> - We have addressed the reviewer's concern regarding fine-grained control over video content.
> - We have addressed the reviewer's misunderstanding regarding the role/use of LLM in V2P stage. This is designed to be agnostic to the choice of LLM backbone.
> - We have addressed the use of the term ‘user-friendliness’ in RACCooN. We use the term to refer to RACCooN's ability to simplify the video editing process with minimal user effort.
> - We have demonstrated that RACCooN outperforms strong instructional image/video editing models with direct visual editing (MagicBrush / InstructVid2Vid) on several editing skills.
> - We have provided additional discussion regarding the potential benefit of RACCooN in further streamlining.
>
> ---
>
> Thank you for your time and effort in reviewing our paper and for your constructive feedback, which has significantly contributed to improving our work. We hope the added clarifications and the revised submission address your concerns and kindly request to further **reconsider the rating/scoring**. We are happy to provide further details or results if needed.
>
> Warm Regards,
> Authors

---

> ### Author Response · Authors · 2024-12-01
> **We have less than two days left in the discussion period.**
>
> Dear Reviewer h1ZE,
>
> We sincerely appreciate your efforts in reviewing our paper and your constructive comments. Since there are less than two days left in the discussion period, could you please read our responses to check if your concerns are clearly addressed? We believe that our responses resolved all your concerns.
>
> We understand that the criteria for rating a paper can sometimes be subjective; however, if you agree that our work does not have remaining major concerns, we would like to kindly suggest your re-evaluation of the initial rating of this submission.
>
> Please let us know if you have any remaining questions, and we will be more than happy to address them.
>
> Best,
> Authors

---

### Official Review · Reviewer_vHvK · 2024-11-04

**Soundness:** 3
**Presentation:** 3
**Contribution:** 3
**Rating:** 6
**Confidence:** 3

**Summary:**

This paper targets video editing by proposing a video-to-paragraph-to-video pipeline. The editing is achieved based on the modification of the paragraph.
The contribution lies in (1) a novel video editing pipeline, and the editing is based on detailed text descriptions, which is user-friendly. (2) improved detailed video captioning performance by a novel multi-granular pooling. (3) A new dataset with high-detailed captions and object caption-mask pairs to facilitate this framework.

**Strengths:**

- The pipeline for video editing is novel and intuitive. The idea of leveraging the recent development of MLLMs to tackle video editing tasks is interesting.
- The experiments are comprehensive. The results of single object prediction are good and outperform some strong baselines. The results of video editing show remarkable improvement.
- The source code is provided in the supplementary materials.
- The method can be integrated with an inversion-based video editing method, and the results are provided.
- Limitations are discussed.

**Weaknesses:**

The author can consider providing more video results to demonstrate the actual editing performance in different settings. For example, the video comparison results with Fatezero and Token flow. Also the results of the proposed method with VideoCrafter and DynamiCrafter.

**Questions:**

The text encoder used for P2V is Clip? Does the max token length be smaller than the input token length? Since the paragraph description is very long, it may exceed the maximum token length of the text encoder. If so, will the information lost will affect the generation sematics?

---

> ### Author Response · Authors · 2024-11-21
>
> Dear Reviewer ```vHvK```,
>
>
> Thank you for your review and constructive comments. During the rebuttal period, we have made every effort to address your concerns. The detailed responses are below:
>
> > W1: Additional visualization results of video editing and video generation compared to Fatezero/Tokenflow and VideoCrafter/DynamiCrafter, respectively.
>
>
> Thank you for your suggestion. We also agree that providing video qualitative results of ```Tables 5 and 6``` will be meaningful to further highlight the robust capability of RACCooN in different settings, such as video editing and conditional video generation.
>
>
> During the rebuttal period, we have **added qualitative video results of Tokenflow (video editing) and VideoCrafter (conditional video generation) and those with RACCooN (Ours)**, which are included in ```Figures 23 and 24``` in the revision.
>
> As illustrated in ```Figure 23 Left```, we contribute these improvements to our generated captions capturing better multi-granular details for a video than the human ones, e.g., *yellow bag* and *a bench* are missed in human captions.
> It helps align the video with a more detailed text description and also leads to more accurate text-to-visual attention in the inversion process for more targeted editing. Furthermore, we perform video editing with superpixel mask guidance in the diffusion steps. The superpixel mask further boosts accurate video editing by preventing details in regions of no interest and forcing the model to edit the region of interest. Using **a simple human caption** resulted in the distortion of a bench behind the women, whereas our method, supported by detailed captions and superpixel masks, preserved the bench's details. It demonstrates the potential of combining detailed automated captions with mask guidance to improve localized video editing. This is also similarly shown in ```Figure 23 Right```; buildings in the background can be preserved when the style change of a man.
>
> We also demonstrate the significance of our proposed RACCooN framework in **conditional video generation** in a qualitative manner. As illustrated in ```Figure 24```, we visualize single- and multi-scene video generation examples of our RACCooN framework.
> We utilize the same keyframe and generate videos with both ground truth and our generated captions. We observe that our generated detailed captions bring better object structure and motion consistency, while the ground truth one leads the generated horse distorted (See ```Figure 24 (a)```).
>
> In addition, we emphasize that our RACCooN facilitates multi-scene video generation by leveraging automatically generated video descriptions containing multiple sentences depicting different contexts. Given the input video, we select a few keyframes for each sentence based on CLIP similarity and create a seamless single video by composing videos generated from these keyframe-sentence pairs. For TokenFlow, we select top-k relevant frames with the ground truth caption. As shown in ```Figure 24 (b)```, RACCooN efficiently generates new composite video content featuring dynamic scenes, whereas the baseline struggles with providing rich visuality. This discrepancy arises because our proposed framework does not rely on ground-truth prompts, which often present incomplete narratives and lack details, impeding the creation of quality videos. Conversely, our RACCooN automatically generates detailed video descriptions, empowering the video generation models to faithfully represent the desired visual and textual subjects.
>
> ---
>
> > Q1: CLIP is used for RACCooN’s text encoder? what is the max token length?
>
>
> Yes, the RACCooN video diffusion framework is built upon Stable Diffusion 2.0, which uses CLIP-Large as its text encoder. The maximum token size of CLIP is 77, and tokens exceeding this limit will be truncated. However, we note that this token limit is sufficient for most object-level descriptions in our benchmarks, and only a minimal number of cases might exceed this limit, with negligible impact on the overall performance.
>
> As shown in our statistics from the VPLM dataset below, the majority of scenarios, including the examples in ```Figure 5```, comply with the maximum token length. Furthermore, we acknowledge that potential advancements in framework design, such as improved text encoders processing longer context lengths with more advanced diffusion frameworks, could further enhance the capabilities of RACCooN. We believe this is a promising direction for future research, extending RACCooN into a more robust and general-purpose video generation and editing framework with long and substantially detailed instructions/planning.
>
> **Table: stats of VPLM**
> |# object description |  avg. # word | Max # words | # example > 77 words|
> |--|--|--|--|
> |5519   |  36.97 | 87 | 9|

---

> ### Author Response · Authors · 2024-11-25
> **A gentle reminder**
>
> Dear Reviewer ```vHvK```,
>
> Thank you for your effort in reviewing our paper. We kindly notify you that the end of the discussion stage is approaching. Could you please read our responses to check if your concerns are clearly addressed? During the rebuttal period, we made every effort to address your concerns faithfully:
>
> - We have provided additional visualization results of video editing and video generation compared to Tokenflow and VideoCrafter, respectively. Please see our revised submission.
> - We have addressed the question of Reviewer vHvK regarding RACCooN’s text encoder.
>
> Thank you for your time and effort in reviewing our paper and for your constructive feedback, which has significantly contributed to improving our work. We hope the added clarifications and the revised submission address your concerns and kindly request to further **reconsider the rating/scoring**. We are happy to provide further details or results if needed.
>
> Warm Regards,
> Authors

---

> ### Author Response · Authors · 2024-12-01
> **We have less than two days left in the discussion period.**
>
> Dear Reviewer vHvK,
>
> We sincerely appreciate your efforts in reviewing our paper and your constructive comments. Since there are less than two days left in the discussion period, could you please read our responses to check if your concerns are clearly addressed? We believe that our responses resolved all your concerns.
>
> We understand that the criteria for rating a paper can sometimes be subjective; however, if you agree that our work does not have remaining major concerns, we would like to kindly suggest your re-evaluation of the initial rating of this submission.
>
> Please let us know if you have any remaining questions, and we will be more than happy to address them.
>
> Best,
> Authors

---

### Author Response · Authors · 2024-11-21
**General Comment by Authors**

We thank the reviewers for their time and valuable comments. We appreciate the reviewers recognized:
- The pipeline for video editing is **novel and intuitive** ```[vHvK]```
- **Interesting idea of the use of MLLM** to tackle video editing ```[vHvK]```
- The proposed Video-to-Paragraph (V2P) method serves as an **effective video captioner**, capable of capturing multi-granular spatiotemporal features. ```[h1ZE]```
- **Solid and extensive experiments** with **SoTA** performance in video editing tasks ```[vHvK, h1ZE, YJ8i, Bguv]```
- The **expandability and generalizability** of the RACCooN framework, capable of integration with other video inpainting/generation backbones ```[vHvk]```
- The paper is **well-written** and easy to understand, and the **research questions are quite interesting**. ```[YJ8i]```
-  **A novel dataset** is introduced, which can be used as a benchmark for video editing ```[h1ZE]```
- The source code is provided in the supplementary materials. ```[vHvK]```

---
We made every effort to address all the reviewers' concerns faithfully. We provided additional experimental results/analyses and discussions, as requested by the reviewers.
- **Additional visualization results** of video editing and video generation compared to Tokenflow and VideoCrafter ```[vHvK]```
- **Additional comparison with InstructVid2Vid** - video editing methods with natural language instructions ```[Bguv]```

---

During the rebuttal period, we made every effort to address all the reviewers' concerns faithfully.

We hope that our responses have addressed your comments.

We thank you again for reviewing our work.

---

### Meta-Review · Area_Chair_BbvZ · 2024-12-18

**Metareview:**

This paper introduces a novel video-to-paragraph-to-video pipeline for video editing, where videos are first converted into detailed textual descriptions. Users can edit these descriptions to modify video content, which is then used to regenerate the edited video. The authors further propose a multi-granular pooling strategy to enhance the quality of descriptions and introduce a new dataset containing detailed captions. Empirical results demonstrate the approach’s effectiveness for object insertion, removal, and modification.

Despite its novelty, the approach has significant limitations. The reliance on generated descriptions can hinder applicability, particularly if required details are omitted in the descriptions. Reviewers also question the necessity of using detailed captions as intermediate representations, where this choice is less intuitive for users and lacks clear justification.

The AC agrees with the reviewers that the need and benefits of using detailed captions as intermediate representations are not clearly described. The authors’ rebuttal fails to provide sufficient evidence to justify this approach or to show why it is superior. Since this is the paper's main claim and contribution, the lack of a strong justification undermines the contribution of the work. Therefore, the AC recommends rejection.

**Additional Comments On Reviewer Discussion:**

The authors argue that the claim that the proposed approach relies on accurate V2P is a misunderstanding, but they do not provide a clear explanation for the argument. While the authors claim that the proposed method can still work if the V2P module fails to describe the editing target, this claim is not backed by concrete evidence and remains unconvincing. Similarly, the rebuttal addressing concerns about the method being less intuitive for users lacks supporting evidence and is therefore unpersuasive. Overall, the AC finds the authors’ responses regarding the necessity and importance of detailed text descriptions for editing to be superficial and lacking solid justification, leaving the design choice of using detailed text descriptions as an intermediate representation unjustified.

---

### Decision · Program_Chairs · 2025-01-22

Reject